# Hydrovoltaic effect-enhanced photocatalysis by polyacrylic acid/cobaltous oxide–nitrogen doped carbon system for efficient photocatalytic water splitting

Xu Xin[1], Youzi Zhang [ID][1], Ruiling Wang[1], Yijin Wang[1], Peng Guo[1] & Xuanhua Li [ID][1] ✉

Severe carrier recombination and the slow kinetics of water splitting for photocatalysts hamper their efficient application. Herein, we propose a hydrovoltaic effect-enhanced photocatalytic system in which polyacrylic acid (PAA) and cobaltous oxide (CoO)–nitrogen doped carbon (NC) achieve an enhanced hydrovoltaic effect and CoO–NC acts as a photocatalyst to generate $H_2$ and $H_2O_2$ products simultaneously. In this system, called PAA/CoO–NC, the Schottky barrier height between CoO and the NC interface decreases by 33% due to the hydrovoltaic effect. Moreover, the hydrovoltaic effect induced by $H^+$ carrier diffusion in the system generates a strong interaction between $H^+$ ions and the reaction centers of PAA/CoO–NC, improving the kinetics of water splitting in electron transport and species reaction. PAA/CoO–NC exhibits excellent photocatalytic performance, with $H_2$ and $H_2O_2$ production rates of 48.4 and 20.4 $mmol\,g^{-1}\,h^{-1}$, respectively, paving a new way for efficient photocatalyst system construction.

Hydrogen ($H_2$) and hydrogen peroxide ($H_2O_2$) are important chemicals in the energy and chemical industries. Photocatalysis is an appealing route toward a sustainable future because it converts solar energy into $H_2$ and $H_2O_2$ from earth-abundant water ($H_2O$) in an ecologically friendly manner. Although considerable effort has been exerted to improve photocatalytic performance, such as defect engineering[1–4], cocatalyst decoration[5,6], and heterojunction construction[7–10], the solar energy conversion efficiency of photocatalysts remains low. Severe carrier recombination and the slow kinetics of the interfacial electron transport of photocatalysts hinder efficient photocatalysis.

External fields, such as thermal, electric, microwave, ultrasonic, and magnetic fields, have been used recently to improve photocatalytic performance significantly[11]. Hydrovoltaic technology is a renewable energy harvesting method that can directly generate electricity by nanostructured materials interacting with water[12–22]. Researchers have designed hydroelectric generators to supply power for various applications[23–32]. For example, a hydroelectric generator

comprising the ionic polymer Nafion and a poly(N-iso-propylacrylamide) hydrogel was developed to generate electricity[33]. A metal–organic framework–based nanomaterial made of UiO-66 nanoparticles grown on two-dimensional AlOOH nanoflakes was designed for small electric appliances[25]. However, an integrated hydrovoltaic effect-enhanced photocatalytic system for water splitting has not been designed. In general, a hydroelectric generator spontaneously absorbs water as it approaches water molecules; then, hydration occurs and induces the formation of large numbers of free charge carriers ($H^+$ or $OH^-$ ions) via ionization[26–29]. These carriers then diffuse with the aid of moisture on the nanostructured materials, giving rise to electric potential generation[14,30–36]. The above characteristics are crucial for promoting charge separation and improving photocatalytic kinetic reactions, but they have not been used in photocatalytic water splitting.

Here, we present a hydrovoltaic effect–enhanced photocatalytic system in which polyacrylic acid (PAA) and cobaltous oxide

[1]State Key Laboratory of Solidification Processing, Center for Nano Energy Materials, School of Materials Science and Engineering, Northwestern Polytechnic University, Xi'an 710072, China. ✉e-mail: lixh32@nwpu.edu.cn

(CoO)−nitrogen doped carbon (NC) synergistically achieve an enhanced hydrovoltaic effect and CoO−NC produces $H_2$ and $H_2O_2$ during photocatalytic water splitting. The proposed system is named PAA/CoO−NC. The height of the Schottky barrier between CoO and the NC interface decreases by 33% due to the hydrovoltaic effect, thereby improving charge separation and transfer. The hydrovoltaic effect also induces a strong interaction between diffuse $H^+$ carriers and the reaction centers of PAA/CoO−NC, which improves the kinetics of water splitting. These features collectively enhance the photocatalytic performance of the proposed system. PAA/CoO−NC exhibits an apparent quantum yield (AQY) for $H_2$ production of 56.2% at 400 nm, and high evolution rates of 48.4 and 20.4 mmol $g^{-1}$ $h^{-1}$ for $H_2$ and $H_2O_2$, respectively, outperforming analog photocatalysts.

## Results
### Synthesis and characterization
A hydrovoltaic effect-enhanced photocatalytic system composed of polyacrylic acid (PAA) and cobaltous oxide (CoO)−nitrogen doped carbon (NC), named PAA/CoO−NC, was prepared as shown in Fig. 1a. CoO−NC, which was fabricated via thermal treatment, was derived from a Co-based metal−organic framework and then dispersed in a solution containing an acrylic monomer (AA) and ammonium persulfate (APS) for cross-linking. Afterward, CoO−NC was encapsulated in the cross-linked PAA to form PAA/CoO−NC. Figure 1b shows a schematic diagram of the hydroelectric generator of PAA/CoO−NC, and its fabrication process of casting film is shown in detail in Fig. S1. A photo of a typical PAA/CoO−NC hydroelectric generator is displayed in Fig. 1c. Scanning electron microscopy (SEM) and transmission electron microscopy (TEM) measurements were conducted to characterize material morphology. The SEM image of PAA/CoO−NC confirms that its porous structure acts as a conduit for water transportation (Fig. 1d). The TEM image of the proposed system shows cross-linked PAA chains wrapping CoO−NC nanoparticles (Fig. 1e). The high-resolution TEM (HRTEM) image of PAA/CoO−NC shows a 0.28 nm lattice of the (200) plane of CoO[11]. The surrounding amorphous substance is attributed to NC and PAA (Fig. 1f). CoO−NC has a polyhedral morphology, as shown in Fig. S2a and S2b. The SEM image of the CoO−NC casting film shows a nanoparticle packing morphology (Fig. S2c). The high-angle annular dark-field scanning TEM (HAADF-STEM) elemental mapping images of CoO−NC and PAA/CoO−NC both show the distributions of the Co, O, C, and N elements (Fig. S2d and S2e). Cross-sectional images of both CoO−NC and PAA/CoO−NC show a thickness of 300 μm (Fig. S2f and S2g).

The X-ray diffraction (XRD) spectra of PAA/CoO−NC and CoO−NC display characteristic peaks corresponding to the face-centered cubic structure of CoO (JCPDS 48-1719) (Fig. 1g)[8,11]. The zeta potential measurement in water confirms the negatively charged surface of PAA/CoO−NC; it has a potential of −23 mV, which can attract counterions of $H^+$ (Fig. 1h)[23]. Fourier transform infrared (FTIR) spectroscopy shows the

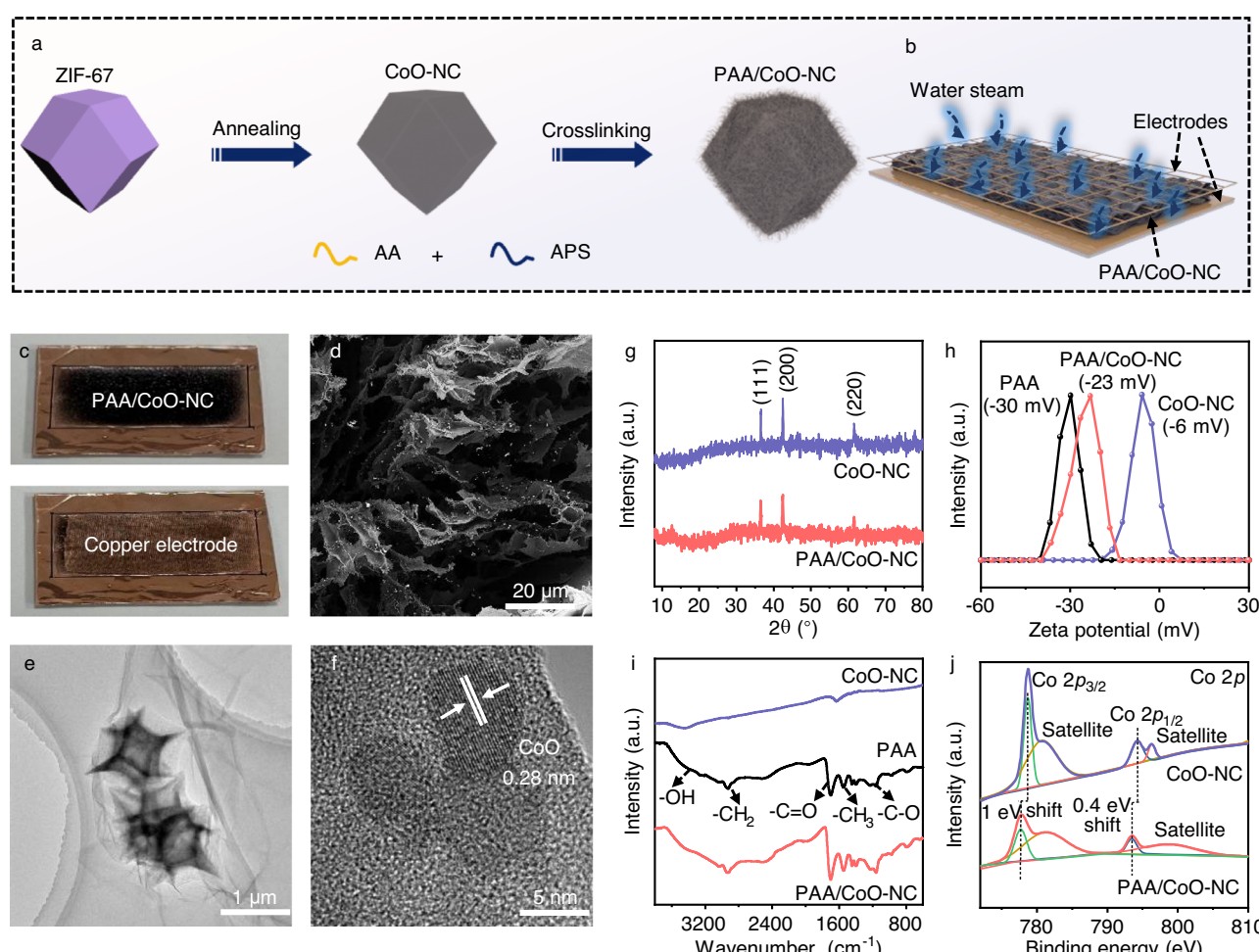

**Fig. 1 | Synthesis and characterization of the hydrovoltaic effect-enhanced photocatalysis system. a** The prepared process of PAA/CoO−NC (AA: acrylic acid; APS: ammonium persulfate); **b** PAA/CoO−NC hydroelectric generator preparation (see Fig. S1 for the preparation process for details); **c** Photo of the prepared PAA/CoO−NC hydroelectric generator; **d** SEM image of PAA/CoO−NC; **e** TEM image of PAA/CoO−NC; **f** HRTEM image of PAA/CoO−NC; **g** XRD patterns of CoO−NC and PAA/CoO−NC; **h** Measured zeta potentials of CoO−NC, PAA and PAA/CoO−NC; **i** FTIR spectra of CoO−NC, PAA and PAA/CoO−NC; **j** Co 2*p* XPS spectra of CoO−NC and PAA/CoO−NC. Source data are provided as a Source Data file.

primary stretches of −OH, −C = O, −C–O, −CH$_2$, and −CH$_3$ in PAA/
CoO−NC due to the introduced PAA (Fig. 1i)[9]. X-ray photoelectron
spectroscopy (XPS) measurements are shown in Fig. 1j and S3. The Co
2$p$ XPS spectra of PAA/CoO−NC and CoO−NC show 2$p_{3/2}$ and 2$p_{1/2}$
features and two satellite peaks, confirming the CoO chemical
nature[8,11]. The slight energy shifts of Co 2$p_{3/2}$ and 2$p_{1/2}$ in PAA/CoO−NC
are due to the strong interaction between CoO and PAA[12]. The CoO
content percentage in CoO−NC and PAA/CoO−NC is about 41% and
23%, respectively (Tables S1–S3). The C 1$s$ and N 1$s$ XPS spectra in
Fig. S3a and S3b, respectively, show the characteristic C–C, C = N,
C = O, and Co–N peaks for PAA/CoO−NC and CoO−NC, verifying the
formation of the NC structure[8]. The electron spin resonance peak at
$g = 2.004$ denotes the CoO oxygen defect observed in CoO−NC and
PAA/CoO−NC (Fig. S4)[7]. PAA/CoO−NC is more hydrophilic than
CoO−NC due to the oxygen-containing functional groups of PAA,
as revealed by the water contact angle test (Fig. S5). The
Brunauer–Emmett–Teller (BET) measurement shows a large specific
surface area of 102 m$^2$ g$^{-1}$ and an average mesopore diameter of 20 nm
for PAA/CoO−NC (Fig. S6).

## Hydrovoltaic effect and voltage tuning

The hydrovoltaic process and voltage generation are highly influenced
by the ion diffusion distance, moving velocity, and wetting behavior, as
shown in Eq. (1)[12]:

$$
\begin{aligned}
V &= \sum_{m=1}^{n} i_m \Delta R_m \\
&= \int_0^d \frac{x}{d} q_0 W v \frac{R_{sq}}{W} dx \\
&= \frac{1}{2} R_{sq} q_0 v d
\end{aligned}
\tag{1}
$$

where $V$, $W$, $i_m$, and $\Delta R_m$ are the voltage, width, current, and resistance,
respectively; $x$, $d$, $q_0$, $v$, and $R_{sq}$ denote the water diffusion distance,
water diffusion depth, charge amount, moving velocity, and square
resistance, respectively. The wetting behavior of the electricity gen-
erator can be regulated by the relative humidity (RH) in the system.
The moving velocity of water in the electricity generator can be
explored by illuminating the system with light[12]. These processes are
further explained by the ionization balance and diffusion of water
molecules, as expressed by the following van't Hoff Eq. (2) and Nernst
Eq. (3)[12,26–31]:

$$
\frac{d \ln k}{dT} = \frac{\Delta_r H_m^*}{RT^2}
\tag{2}
$$

$$
V = \left| \frac{RT}{F} \ln \frac{a_2}{a_1} \right|
\tag{3}
$$

where $k$, $T$, R, $\Delta_r H_m^*$, F, and $V$ denote the balance coefficient, tem-
perature, gas constant, standard enthalpy change, Faraday constant
and voltage, respectively, and $a_2$ and $a_1$ represent the high- and low-
concentration H$^+$ ion activities, respectively. According to Eqs. (2) and
(3), at ambient temperature, voltage is proportional to the RH and
moving velocity, which increases the probability of ionization reaction
(i.e., provides sufficient water molecules to enable mobility of H$^+$ ions)
and ion diffusion coefficient (i.e., induces more H$^+$ ions to diffuse)[31,37].

Based on the above theory, we practically explored the factors of
voltage tuning in the hydrovoltaic effect-enhanced photocatalytic
system. First, we optimized the surface area, and thickness of the
electricity generator, as illustrated in Fig. S7, and Fig. 8 and Note S1.
The optimal surface area, and thickness of the electricity generator
are 15 cm$^2$, and 300 μm, respectively. We tuned the RH conditions at
different rates of Ar/H$_2$O steam injection into the reactor
(100–1300 ml h$^{-1}$), as shown in Fig. 2a. PAA/CoO−NC reaches the

highest voltage (~280 mV) at a 1100 ml h$^{-1}$ Ar/H$_2$O steam injection
rate, suggesting a strong correlation between the electrical genera-
tion performance and moisture content variation. At higher humidity
(Ar/H$_2$O steam injection 1300 ml h$^{-1}$), the water steam will completely
saturate and cover the entire membrane, resulting in the dis-
appearance of water steam gradient and hydrovoltaic effect[16]. The
output voltage of CoO−NC shows a similar variation in the set RH
range (Fig. S9), with an optimized voltage of 106 mV at the 1100 ml h$^{-1}$
Ar/H$_2$O steam injection rate; this figure is lower than that of PAA/
CoO−NC due to CoO−NC's slightly weak hydrovoltaic effect
generation.

Then, we investigated the influence of light illumination on the
system for moisture moving. The light illumination induces a higher
temperature on the surface of PAA/CoO−NC, bringing an increased
moving velocity of water steam in the nanochannel of PAA/CoO−NC
(detailed relation seen in Supporting Information of Note S2, Fig. S10).
In addition to an increased velocity of water steam induced by the light
illumination, the inhomogeneous distribution of the heat (i.e. thermo-
electric effect) and photogenerated carriers (i.e. photoelectric effect)
induced by light illumination have effects on an enhanced voltage
generation. PAA/CoO−NC, PAA, and CoO−NC exhibit voltages of 402,
86, and 230 mV, respectively, under light illumination (Fig. 2b); these
results exceed those in the case without light illumination, as shown in
Fig. S11. Thus, light illumination can increase the moving velocity,
thereby enhancing voltage generation. Power generation for PAA/
CoO−NC was investigated by connecting different external load
resistances (from 1 Ω to 150 kΩ) to the system (Fig. 2c). At a load
resistance of 33 kΩ, the power density reaches maximum values of 0.81
and 0.49 μW cm$^{-2}$ for the cases with and without light illumination,
respectively, whereas both current densities gradually decrease to
zero. Notably, the optimized electricity generation performance of
PAA/CoO−NC is comparable to that of hybrid nanomaterials in
moisture-induced electricity generation (Note S3 and Table S5). The
power generation for CoO−NC shows a similar variation but with
slightly lower power densities of 0.33 and 0.19 μW cm$^{-2}$ for the cases
with and without light illumination, respectively (Fig. S12).

Figure 2d shows a photo of PAA/CoO−NC for electricity genera-
tion with light illumination (AM 1.5 G, 100 mW cm$^{-2}$) at the Ar/H$_2$O
steam injection rate of 1100 ml h$^{-1}$; the findings verify the presence of
the hydrovoltaic effect in the proposed system. The PAA/CoO−NC
hydroelectric generator has been successfully constructed based on a
moisture electrokinetic effect in a nanochannel of PAA/CoO−NC[15,30].
The water steam is exposed on the surface of PAA/CoO−NC and then
diffused along the nanochannel to the bottom (the detail mechanism
and water steam diffusion path seen in Fig. S13). The PAA/CoO-NC
generates a pressure-driven flow carriers counter-ions of H$^+$ to form an
electric current in the flow, eventually reaching an equilibrium of H$^+$
ions diffusion, resulting in a constant voltage output[15,30]. We further
investigated the hydrovoltaic electrical field and H$^+$ ion distribution
using the Gouy–Chapman–Stern model, as shown in Fig. 2e[11,26,38]. Note
S4 contains the governing equations and boundary conditions used in
the model. The simulated voltage for PAA/CoO−NC is optimally
400 mV, which is close to the experimental value in Fig. 2b. The H$^+$ ion
distribution varies across the material and shows a spike adjacent to
where the steam flows (upper portion)[39,40]. The simulated results of
CoO−NC under identical conditions show a voltage of ~200 mV, as
shown in Fig. S14; this result is consistent with the experimental value
of CoO−NC in Fig. 2b.

## Photocatalytic water splitting performance

We conducted hydrogen production experiments to confirm the
hydrovoltaic electric field driven electrochemical reactions during
photocatalytic water splitting. The ultraviolet−visible diffuse
reflectance spectroscopy (UV−vis DRS) measurements of PAA/
CoO−NC and CoO−NC exhibit similar optical absorption results

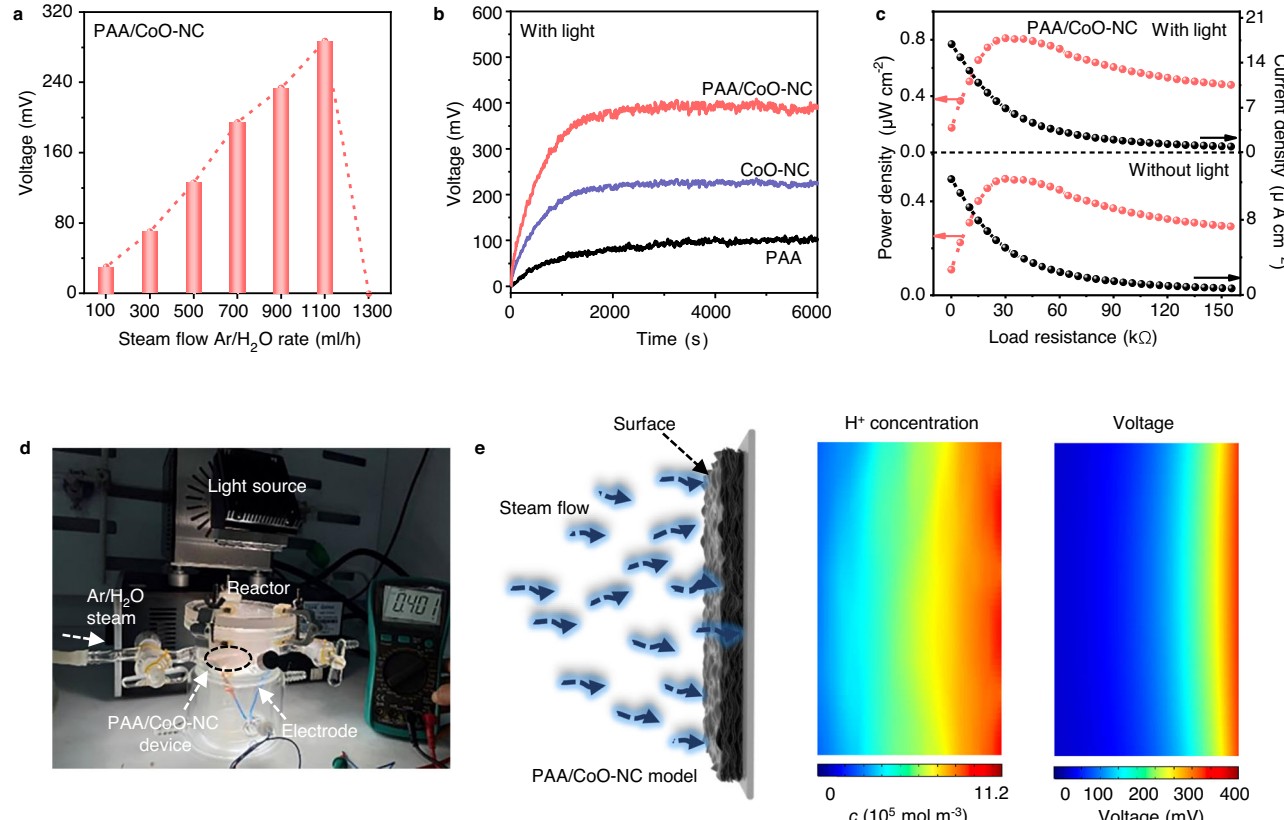

**Fig. 2 | Hydrovoltaic voltage generation and factors tuning of the hydrovoltaic effect-enhanced photocatalysis system. a** Measured output voltage of PAA/CoO−NC under different Ar/$H_2$O flow rates; **b** Measured output voltage for CoO−NC, PAA and PAA/CoO−NC over time with light illumination (light intensity: AM 1.5 G; 100 mW cm$^{-2}$); **c** Output power and current density of PAA/CoO−NC under different load resistances with or without light illumination. Red arrow represents power density; Black arrow represents current density; **d** Photo of the PAA/CoO−NC hydroelectric generator for electricity generation under light illumination (AM 1.5 G, 100 mW cm$^{-2}$) and at Ar/$H_2$O injection rate of 1100 ml h$^{-1}$; **e** Numerical simulations of the electric field and $H^+$ ion distribution on PAA/CoO−NC in cross sectional view (water steam diffuses from the surface to the bottom of PAA/CoO−NC). Source data are provided as a Source Data file.

(Fig. S15). The photocatalytic performance of the system was investigated at different Ar/$H_2$O steam injection rates (100–1300 ml h$^{-1}$) according to various ambient RH values in the reactor. A 0.1 wt% Pt cocatalyst was loaded on the photocatalyst via photodeposition. The $H_2$ production amount of PAA/CoO−NC gradually improves from 3.1 to 12.1 mmol in 5 h as the Ar/$H_2$O steam injection rate increases from 100 to 1100 ml h$^{-1}$ (Fig. 3a). This pattern is consistent with the increasing hydrovoltaic electric field in Fig. 2a. $H_2$ production drops to 2.4 mmol at an Ar/$H_2$O steam injection rate of 1300 ml h$^{-1}$ due to the disappearance of the hydrovoltaic effect, close to the value of 2.2 mmol submerged in water. Accordingly, the main oxidation product of $H_2O_2$ is detected (Note S5 and Fig. 3b); it increases from 2.5 to 5.1 mmol and then decreases to 2.1 mmol. This variation trend is similar to that of $H_2$ production. The optimized $H_2$ evolution rate of PAA/CoO−CN is 48.4 mmol g$^{-1}$ h$^{-1}$ at the Ar/$H_2$O steam injection rate of 1100 ml h$^{-1}$, as depicted in Fig. 3c. This result is approximately five times higher than that at the 1300 ml h$^{-1}$ Ar/$H_2$O steam injection rate (9.8 mmol g$^{-1}$ h$^{-1}$). The corresponding $H_2O_2$ production rate is 20.4 mmol g$^{-1}$ h$^{-1}$, which is more than twice that at the 1300 ml h$^{-1}$ Ar/$H_2$O steam injection rate (8.4 mmol g$^{-1}$ h$^{-1}$). The photocatalytic performance of CoO−NC, as shown in Fig. S16 further confirms the enhanced function of its hydrovoltaic effect.

The AQY for $H_2$ evolution was measured at the 1100 ml h$^{-1}$ Ar/$H_2$O steam injection rate using various band-pass filters to provide monochromatic light. The AQY for $H_2$ production at 400 nm is 56.2% (Fig. 3d and Note S6), which is higher than the measured AQY values of 11.7% for CoO−NC and 5.8% for the PAA/CoO−NC submerged in water

(Fig. S17). The enhanced photocatalytic performance of PAA/CoO−NC is the highest in the comparison of the Co-based photocatalysts (Fig. 3e and Table S6)[41–53]. In addition, the performance of a general photocatalytic system in bulk water degrades to 84%, whereas no noticeable degradation after four cycles of 20 h reaction in the hydrovoltaic-enhanced photocatalytic system (Fig. S18). After a longer reaction period of 80 h, the performance retained 92% of its initial activity for PAA/CoO−NC, indicating a stable structure for hydrovoltaic-enhanced water splitting system (Fig. S19). In addition, the hydrovoltaic effect generation and its enhanced photocatalysis were also demonstrated in a natural water-evaporation-induced hydrovoltaic system (Fig. S20). The PAA/TiO$_2$−NC photocatalyst has also prepared to further demonstrate the generality of hydrovoltaic-enhanced photocatalysis as shown in Fig. S21.

## Kinetics of hydrovoltaic-enhanced water splitting
We studied the in situ Raman spectra of PAA/CoO−NC at the 1100 ml h$^{-1}$ Ar/$H_2$O steam injection rate to better understand the hydrovoltaic effect-enhanced photocatalysis from the perspectives of hydrogen binding energy and water structure evolution. The position and intensity of the active hydrogen adsorption peak strongly depend on the hydrogen binding energy characteristics, which are associated with the potential field intensity and water interaction during the electrochemical reaction[54–57]. We loaded minor Pt nanoparticles to detect the Raman signal of Pt-H as a descriptor for hydrogen binding energy[58]. A peak centered at 2100 cm$^{-1}$, assigned to Pt-H vibration, is detected in Fig. 4a in the in situ Raman spectra of PAA/CoO−NC under light illumination[59],

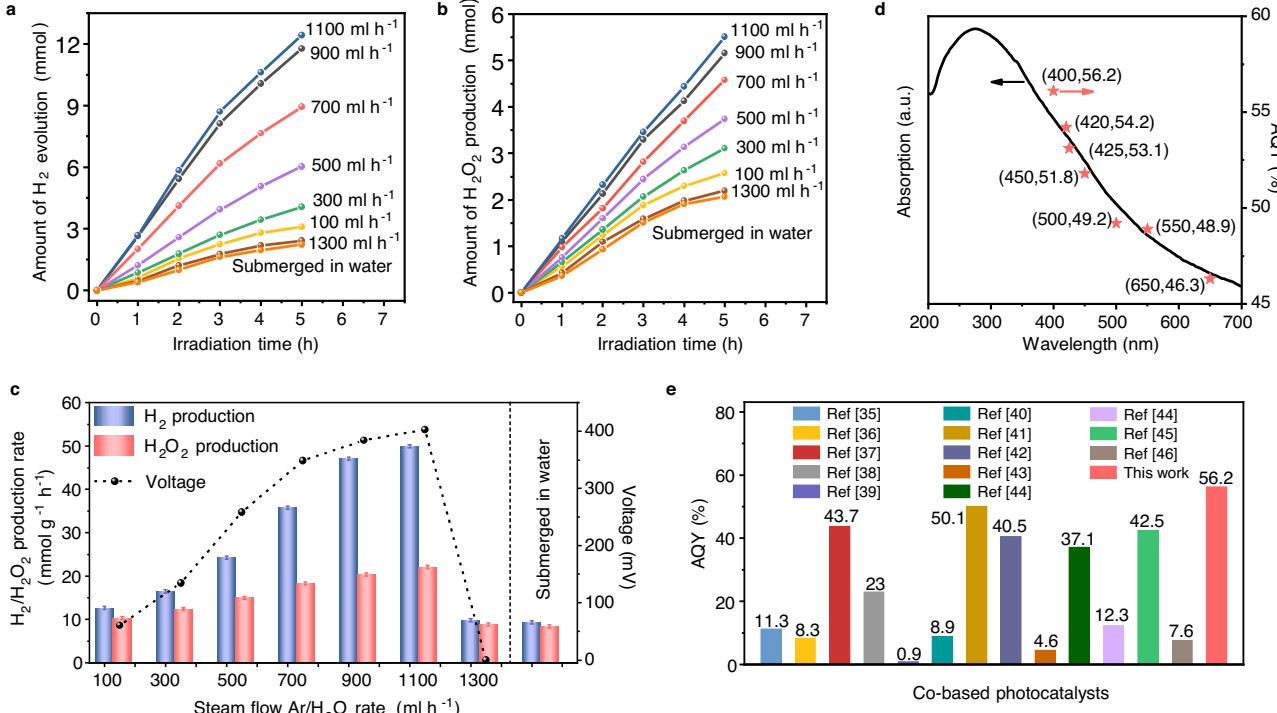

**Fig. 3 | Photocatalytic water splitting performance of the hydrovoltaic effect-enhanced photocatalysis system. a** Time-dependent photocatalytic $H_2$ production of PAA/CoO–NC at different $Ar/H_2O$ rates from 100 to 1300 ml h$^{-1}$, and submerged in water; Pt cocatalyst is loaded using a photodeposition method; The light source is a solar simulator at AM 1.5 G illumination (100 mW cm$^{-2}$); **b** Time-dependent photocatalytic $H_2O_2$ production of PAA/CoO–NC at different $Ar/H_2O$ rates from 100 to 1300 ml h$^{-1}$, and submerged in water; **c** The photocatalytic $H_2$/ $H_2O_2$ production rates of PAA/CoO–NC at different $Ar/H_2O$ rates from 100 to 1300 ml h$^{-1}$ corresponding to the voltage variation, and submerged in water. Error bars represent the standard deviations from the statistic results of three sets of experiments; **d** UV–vis DRS spectra and the wavelength-dependent AQY of PAA/ CoO–NC for $H_2$ production. Red arrow represents AQY; Black arrow represents absorption; **e** The comparison of AQY values on various Co–based photocatalysts. Source data are provided as a Source Data file.

which gradually increases in intensity over time. The redshift of the Pt–H peak from 2097 to 2191 cm$^{-1}$ confirms its intrinsic surface-adsorbed species and Stark tuning phenomenon driven by potential field[56,59]. The O–H stretching observed in PAA/CoO–NC evolves at ~3200 and 3400 cm$^{-1}$, which are attributed to tetrahedrally coordinated water (ice-like water) and trihedrally coordinated water (liquid-like water), respectively. The O–H stretching increases in intensity over time, indicating that PAA/CoO–NC interacts strongly with the water molecules[60]. A similar Pt–H peak and O–H vibration are observed in CoO–NC, as shown in Fig. 4b, but its intensity increase slowly and peak shifts reduce, indicating lower catalytic activity.

The two-dimensional contour plots in Fig. 4c, d show that the Pt–H vibrations of PAA/CoO–NC and CoO–NC, respectively, gradually intensify with their hydrovoltaic electric field intensity over time. The binding energy of Pt–H and the peak shift can be linearly correlated based on a simple harmonic motion for the Raman peak (Note S7), as shown in Figs. 4e, f[60]. A slope of 235 cm$^{-1}$ V$^{-1}$ for the Pt–H vibration of PAA/CoO–NC is calculated in a hydrovoltaic potential window of 0–400 mV with an increase in the Raman area ratio (Fig. 4e); this value exceeds that of CoO–NC (170 cm$^{-1}$ V$^{-1}$) (Fig. 4f). The higher slope indicates the greater hydrogen binding energy associated with the high hydrovoltaic electric field provided by PAA/CoO–NC. In comparison, the in situ Raman spectra of the PAA/CoO–NC submerged in water show the smallest intensity of Pt–H vibration and O–H stretching due to the disappearance of the hydrovoltaic effect and bulk water interaction, as shown in Fig. S22. The in situ Raman spectra of PAA/Pt was also investigated in Fig. S23 to exclude the catalytic activity of the substrate, indicating a great enhancement on photocatalyst induced by hydrovoltaic effect.

## Hydrovoltaic-enhanced photocatalytic mechanism

The properties of the potential barrier with the hydrovoltaic effect were evaluated based on the thermionic emission model and theory[61]. According to temperature-dependent $I-V$ characteristic measurements at 300–370 K at different applied bias values, the equation is expressed by:

$$\ln\left(-\frac{I}{T^2}\right) = \ln(AA^*) - \frac{q\varnothing_E(V_{app})}{kT} \tag{4}$$

where $V_{app}$ is the applied voltage, $I$ is the current, A is the effective contact area, $A^*$ is the Richardson constant, and $q\varnothing_E(V_{app})$ is the energy barrier height in a voltage-dependent manner (calculation details in Note S8). Figure 5a depicts Arrhenius plots calculated and extracted from the $I-V$ plots (Fig. S24). PAA/CoO–NC exhibits the smallest activation energy for carrier transport, with a value of 15.42 eV, which is favorable for charge transport. The extracted effective energy barrier height of $q\varnothing_E(V_{app})$ is displayed in Fig. 5b. The height of the extracted effective energy barrier around zero bias for PAA/CoO–NC reduces by 33% due to the hydrovoltaic effect. The Electrochemical impedance spectroscopy (EIS) measurements of PAA/ CoO–NC show gradually depressed semicircle diameters with longer water interactions, verifying the improved conductivity of PAA/ CoO–NC (Fig. S25)[9].

A band diagram was created to better understand the dominant interfacial electric field induced by the hydrovoltaic effect. The energy band structures of individual CoO and NC were calculated using Mott–Schottky (M–S) plots and ultraviolet photoelectron spectroscopy (UPS) (Fig. 5c, Fig. S26). The detailed Fermi levels ($E_f$), conduction band minimum (CBM), and valence band maximum

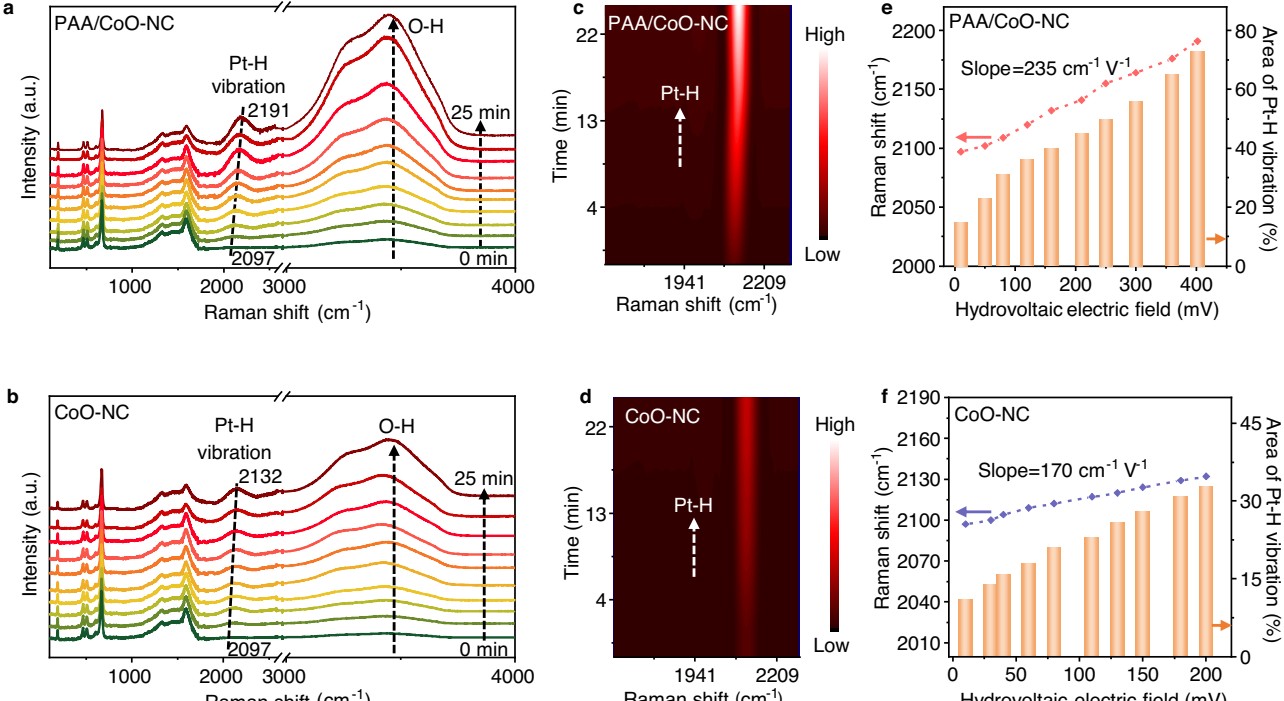

**Fig. 4 | The kinetics of water splitting reaction of the hydrovoltaic effect-enhanced photocatalysis system. a** In situ Raman spectra of the PAA/CoO–NC surface over time with hydrovoltaic effect under light irradiation; **b** In situ Raman spectra of the CoO–NC surface over time with hydrovoltaic effect under light irradiation; **c** 2D contour maps of Pt-H vibrations on PAA/CoO–NC; **d** 2D contour maps of Pt-H vibrations on CoO–NC; **e** Raman shifts and area ratios of the Pt-H bond versus the hydrovoltaic potential range produced by the electric field at the PAA/CoO–NC surface. Red arrow represents Raman shift; Orange arrow represents area of Pt-H vibration (%); **f** Raman shifts and area ratios of the Pt-H bond versus the hydrovoltaic potential range produced by the electric field at the CoO–NC surface. Blue arrow represents Raman shift; Orange arrow represents area of Pt-H vibration (%); For the in situ Raman measurements, Pt cocatalyst is loaded through a photodeposition method; The light source is a solar simulator at AM 1.5 G illumination (100 mW cm$^{-2}$). Source data are provided as a Source Data file.

(VBM) are shown in Fig. 5d. CoO and NC initially have different $E_f$ and surface states. When a CoO–NC contact forms, energy band bending and an internal electric field generate at the interface, resulting in a Schottky barrier and the upward bending of the energy band between CoO and NC. The photogenerated electrons transfer to NC is blocked by the Schottky barrier (Fig. 5e). The hydrovoltaic effect generates a positive interfacial electric potential drop that affects (decreases by 33%) the Schottky barrier height of the catalyst relative to the external potential intensity (Fig. 5f); moreover, the electrons are driven across the photocatalyst, thereby significantly promoting carrier separation and transport[61]. Electron paramagnetic resonance (EPR) experiments were performed to validate the generation of •OH radicals on the samples. A signal of •OH radicals was observed in CoO and PAA/CoO–NC[62]. The high signal intensity of PAA/CoO–NC indicates that the photogenerated holes are driven by the interfacial electric field between CoO and NC and consumed on CoO to generate $H_2O_2$ (Fig. 5g). Density functional theory (DFT) based calculations further indicate that the CoO–NC photocatalyst prefers to generate $H_2O_2$ rather than $O_2$ evolution in the system (Fig. S27).

Based on the energy band structure and the above results, the proposed hydrovoltaic effect-enhanced photocatalysis mechanism on PAA/CoO–NC is depicted in Fig. 5h. Under light illumination, PAA/CoO–NC absorbs light, and the photogenerated carriers are excited and separated. The photoinduced electrons are driven by the hydrovoltaic effect from the CBM of CoO to NC and then transferred to the Pt cocatalyst to produce $H_2$. Photogenerated holes are consumed in the VBM of CoO to generate $H_2O_2$. The height of the Schottky barrier between CoO and the NC interface decreases due to the hydrovoltaic effect, which significantly promotes carrier separation and transfer on the photocatalyst. Moreover, the hydrovoltaic effect generates a strong interaction between the H$^+$ carriers and reaction centers of the nanostructure, thereby further improving the kinetics of water splitting. These features collectively enhance the photocatalytic performance of the proposed system.

## Discussion

We construct a hydrovoltaic effect-enhanced photocatalytic system called PAA/CoO–NC, in which PAA and CoO–NC synergistically achieve an enhanced hydrovoltaic effect and CoO–NC serves as a photocatalyst to produce $H_2$ and $H_2O_2$ simultaneously. The Schottky barrier height between CoO and the NC interface decreases by 33% due to the introduced hydrovoltaic effect. Moreover, a strong interaction between the H$^+$ carriers and reduction reaction centers of the PAA/CoO–NC nanostructure is formed along the water steam diffusion, improving the kinetics of water splitting and electron transport. Therefore, PAA/CoO–NC exhibits superior photocatalytic performance, with $H_2$ and $H_2O_2$ production rates of 48.4 and 20.4 mmol g$^{-1}$ h$^{-1}$, respectively, and a high AQY of 56.2% at 400 nm. Our findings introduce a new way of constructing efficient photocatalyst systems involving the introduction of a hydrovoltaic effect to photocatalytic reactions

## Methods

### Preparation of cobaltous oxide–nitrogen doped carbon (CoO–NC)

To synthesize ZIF-67, a solution of 0.984 g 2-methylimidazole ($C_4H_6N_2$, Aladdin., 98%) in 15 ml of ethanol was added dropwise to a solution of 0.873 g cobalt nitrate hexahydrate ($Co(NO_3)_2$·6H$_2$O, Shanghai Macklin Biochemical Co., Ltd., AR, 99%) and 0.3 g polyvinylpyrrolidone (PVP, Aladdin, molecular weight 10000) in 15 ml of ethanol with vigorous stirring. The molar ratio of the metal salt to

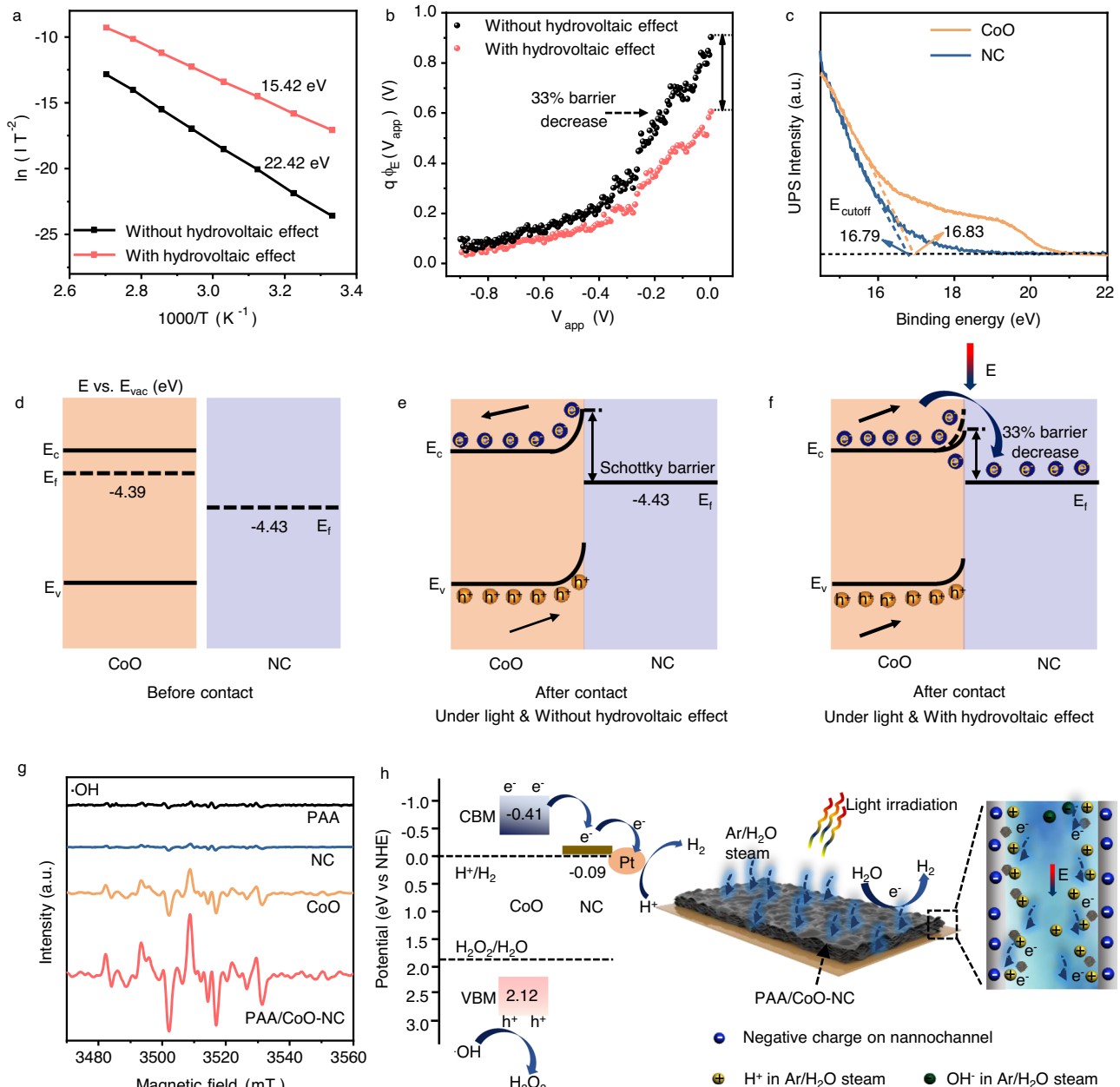

**Fig. 5 | Photocatalytic mechanism of the hydrovoltaic effect-enhanced photocatalysis system. a** The measured Arrhenius plots extracted from *I–V* curves; **b** The extracted effective energy barrier height $q\varnothing_E(V_{app})$ with voltage dependence; Negative bias corresponds to a Schottky contact, indicating the observed large scattering near zero bias, and the data of the scattered point are measurable in the range of applied bias; **c** UPS spectra of the cut-off region (Secondary cut-off binding energy: $E_{cutoff}$) for CoO and NC; **d–f** Band diagram mechanisms for equilibrium in the CoO–NC photocatalyst interface and the Schottky barrier height determination for PAA/CoO–NC; The prepared electrode material was subjected to a two-electrode test system with 1100 ml h$^{-1}$ Ar/H$_2$O injection. Potential (E, eV) vs Vacuum potential ($E_{vac}$, eV). Fermi levels ($E_f$), conduction band minimum ($E_c$), valence band maximum ($E_v$). **d** Before contact; **e** In contact without hydrovoltaic effect; **f** In contact with hydrovoltaic electric field (E); **g** EPR spectra of DMPO−•OH trapped on PAA, CoO and PAA/CoO−NC under light irradiation; **h** The band structure and schematic of hydrovoltaic effect-enhanced photocatalytic water splitting performance (E: hydrovoltaic electric field). Source data are provided as a Source Data file.

ligand was 1:4 or 0.003:0.012 mol. After stirring the mixture, ZIF-67 was obtained by letting it sit at room temperature for 24 h. The resulting ZIF-67 (200 mg) was then treated at 600 °C for 1 h under N$_2$ flow, followed by treatment at 200 °C for 30 min in a muffle furnace. This two-step annealing process ensured successful synthesis of CoO-NC, despite the inevitable reduction in surface area caused by sintering. Annealing at 600 °C in the first step allowed for complete decomposition of ZIF-67 and good crystallinity of Co nanoparticles, while annealing at 200 °C in the second step yielded CoO-NC.

## Preparation of polyacrylic acid/cobaltous oxide−nitrogen doped carbon (PAA/CoO−NC)

PAA-functionalized CoO−NC nanocomposites were synthesized via in situ polymerization. First, 0.05 g CoO−NC was dispersed in a solvent mixture consisting of equal volumes of water and ethanol (100 ml, 1:1) by ultrasonication and stirring continuously for a duration of 1.5 h. Subsequently, 5 ml acrylic acid (AA, Sinopharm Chemicals Reagent Co., Ltd.) and 0.1 g ammonium persulfate (APS, Sinopharm Chemicals Reagent Co., Ltd.) was added to the suspension, which was then agitated for half an hour at ambient temperature under a nitrogen atmosphere.

The product was then heated to 80 °C and reacted for 5 h. After centrifugation, the PAA/CoO-NC nanocomposite was collected, followed by a thorough washing with deionized water and subsequent drying. The resulting product was named PAA/CoO-NC for abbreviation.

## Pt cocatalyst loading

To load Pt on PAA/CoO–NC photocatalyst, a standard photochemical procedure was followed. We first dispersed 50 mg of PAA/CoO–NC nanomaterial and 0.25 ml of Chloroplatinic acid hexahydrate ($H_2PtCl_6 \cdot 6H_2O$, 4 mg ml$^{-1}$, ACS reagent) in a 20 ml aqueous solution of $H_2O$. The mixture was then subjected to light treatment for an hour. After that, we centrifuged the sample and washed it twice with deionized water, followed by free-drying. This process resulted in a Pt content of ~0.1 wt% relative to PAA/CoO-NC.

## Data availability

All data generated in this study are provided in the article and Supplementary Information, and the raw data generated in this study are provided in the Source Data file. Source data are provided with this paper.

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

## Acknowledgements

This research is supported by the National Natural Science Foundation of China (22261142666, 52172237), the Shaanxi Science Fund for Distinguished Young Scholars (2022JC-21), the Research Fund of the State Key Laboratory of Solidification Processing (NPU), China (Grant No. 2021-QZ-02), and the Fundamental Research Funds for the Central Universities (3102019JC005, D5000220033). All fundings are awarded to X. L. We thank the members from the Analytical & Testing Center of Northwestern Polytechnical University for the help of XRD, XPS, and SEM characterization.

## Author contributions

X.X. and X.L. conceived the idea. X.L. supervised the project. X.X. performed the synthesis, characterization, and photocatalysis. Y.Z. and P.G. performed the photo-electrochemical experiments. X.L., R.W. and Y.W. analyzed the data and commented on the manuscript.

## Competing interests
The authors declare no competing interests.
