## [Peer Review File · Nature Communications]

REVIEWER COMMENTS

Reviewer #1 (Remarks to the Author):

This work uses the hydrovoltaic effect to enhance the performance of photocatalytic water splitting. The idea is novel. Also, the system has good performance with H₂ and H₂O₂ production rates of 48.4 and 20.4 mmol g⁻¹ h⁻¹, respectively, and a high AQY of 56.2% at 400 nm, making it among the most efficient photocatalysts. Thus I recommend the acceptance of this manuscript. However, the experiments and results are not clearly presented, and mechanism is not well explained, major revision is needed before publication, as noted below.

1. It is not clear how the generated hydrovoltage can enhance the photocatalytic reaction. The hydrovoltage is established between two electrodes along the film, but in Fig. 5, this voltage seems to be applied to CoO/NC interface, there is no direct relation between them.
2. Why H₂O₂ is generated instead of O₂?
3. What is the direction of steam flow? Based on Fig. 5, the flow is along the width of the film. If this is the case, how hydrovoltage is created because two electrodes are symmetric and there should be no voltage difference.
4. The schematic in Fig. 5 is based on water, but most experiments were performed in steam, will the mechanism stay the same?
5. What is the condition of steam? Temperature? Is there any water droplets? According to the phase diagram (J. Phase Equilib., 24 [3] 212-227 (2003)), if the temperature is high, CoO will convert to Co₃O₄. Thus temperature is important.
6. What determines the flow velocity of steam or water? If it is determined by external pressure, then it should not depend on the illumination as in section 2.2, the authors sated "light illumination can increase the moving velocity".
7. In the stability test, does PAA/CoO-NC stay the same after four cycles of 20 h reaction? What will happen if prolong the reaction time?

Reviewer #2 (Remarks to the Author):

The present manuscript reports on Hydrovoltaic effect-enhanced photocatalysis by PAA/CoO–NC system for efficient photocatalytic water splitting. The materials are tested as photocatalysts for H₂ evolution.

Due to its optical properties, PAA/CoO–NC system is an interesting material to explore for photocatalytic application. The manuscript in my view presents a novelty to be published in Nature communication. However, some critical issues need to be addressed and manuscript revised prior to acceptance.

For these reasons, I recommend the publication of the present manuscript in Nature communication. Major critical points are outlined in the following:

1. The increased photocatalytic performance upon modification with CoO is certainly not a surprise as it serves as a co-catalyst or a trap center for the material prepared. Moreover, the manuscript omits some relevant information and review on hydrovoltaic effect as it related to photocatalysis check this article and reference it. (Zhong, T., Li, H., Zhao, T., Guan, H., Xing, L., & Xue, X. (2020). Self-powered/self-cleaned atmosphere monitoring system from combining hydrovoltaic, gas sensing and photocatalytic effects of TiO₂ nanoparticles. *Journal of Materials Science & Technology*. doi:10.1016/j.jmst.2020.11.002)
2. Carbon is known to possess large surface area, however, the work is silent about the effect of CoO on the surface area of the system.
3. The specific wavelength of the light source used in the photocatalytic generation of H₂ is not state, even though Figure S16 depicts several wavelengths was used to determine AQY.
4. I am surprised that the effect of calcination at 600 oC on the surface of the CoO-NC was not reported, particularly if one considers that CoO-NC was annealed at 600°C for 1 h, which might cause sintering and a consequent reduction of the surface area. In other words, one expects the bare CoO-NC to feature a higher specific surface area.
5. If the DRS UV/VIS of both PAA/CoO and CoO-NC shows similar optical properties as presented in Fig S14. What is responsible for the difference in rate of H₂ evolution in both systems Because the difference is bandgap is not obvious.
6. The mechanism presented in section 2.5 did not explain the pathway to the formation of H₂O₂ because in water splitting H₂ and O₂ are normally the by-product (Theoretically).
7. The XPS results in Figure 1j and S3 did not explain if the CoO is decorated on the surface of the materials or found within the matrix, also the actual amount of the CoO as determined by the XPS is not reported.
8. The conclusion stated that the hydrovoltaic effect improves the hydrogen binding energy in the formation of active hydrogen during photocatalytic water splitting, resulting in excellent charge

separation and transfer and accelerated reaction kinetics. However, the result discussion does not support this claim.

Reviewer #3 (Remarks to the Author):

In this manuscript, Xu et al. proposed a hydrovoltaic effect-enhanced photocatalytic system in which an enhanced hydrovoltaic effect was achieved using polyacrylic acid (PAA) and cobaltous oxide (CoO)–nitrogen doped carbon (NC) and CoO–NC acted as a photocatalyst to generate H₂ and H₂O₂ products simultaneously. The hydrovoltaic effect was shown to decrease the Schottky barrier height between CoO and the NC interface and accelerate the kinetics of water splitting, thereby resulting in photocatalytic performance with high H₂ and H₂O₂ production rates. This work realized a synergetic combination of the conventional photocatalysis and the newly emerged hydrovoltaic effect and may open a new sub-direction for sustainable development of clean energy. However, before I can recommend its publication, the authors have to fully address following issues:

1. In the introduction, “Hydrovoltaic technology is a renewable energy harvesting method that uses moisture interactions with nanostructured carbon materials”. The definition is not accurate. Hydrovoltaic technology can not only harvest energy from ambient moisture, but also from evaporation, raindrops, waves and so on. It is characterized by energy conversation based on the interaction between water and functional materials. The authors should find in which paper this terminology was first proposed.

2. In section 2.2, the influence of light illumination on the moisture moving velocity was investigated. An increased velocity results in an enhanced voltage generation. The author should clarify whether the enhancement is contributed solely by the increased moving velocity, any additional contribution from other factors, such as the photogenerated carriers or inhomogeneous distribution of the heat induced by light?

3. In the system, CoO was used as the active centers for water splitting and was shown to have a desirable band alignment with NC groups. I’m wondering if such a strategy of materials design can be generalized to a range of metal oxides, notably TiO₂ that is mostly used for photocatalysis. The authors are suggested to perform additional experiments to demonstrate the generality.

4. The generated hydrovoltaic electricity is by pumping moisture steam through the materials, which costs extra energy and may result in an overall extremely low efficiency. Actually, hydrovoltaic technology enables electricity harvesting from natural evaporation which does not any extra energy input. In this regard, can the authors employ the evaporation-induced electricity to enhance the photocatalysis?

5. To further verify the photocatalytic effect of PAA/CoO-NC, the authors are suggested to carry out control experiments of photocatalytic H₂/H₂O₂ production and provide in situ Raman spectra of pure PAA membranes loaded with Pt cocatalyst at 1100 ml h⁻¹ Ar/H₂O steam injection.

6. There are some misquotes in the introduction, such as “For example, a hydroelectric generator comprising the ionic polymer Nafion and a poly(N-isopropylacrylamide) hydrogel was developed to generate electricity [17, 18]”. Please check the literature comprehensively.

7. In Figure S1, the description on the device fabrication process in figure legend does not match its schematic illustration. For example “Commercial epoxy slurry is used to paint two “L-shaped” with predesigned dimensions on the substrate (Step 1)”; here, the electrodes should be “L-shaped”, not epoxy.

8. In Note S1, “relative humidity by pumping a mixture of dry and wet nitrogen”, while in the paper, the humidity is controlled by “Ar/H₂O steam injection”.

9. In Figure S7 (c) and Figure S8 (b), “The increase in the current density may be attributed to the increased amount of water moving and diffusion as W increases”. While in general perception, the current increases with the width of the hydroelectric generator, and the current density is nearly constant. Please explain it.

10. In Figure 2a, the authors need to explain why the voltage at a 1300 ml h⁻¹ Ar/H₂O steam injection rate is lower than that at 1100 ml h⁻¹. Does the liquid water cover the PAA/CoO-NC membrane?

11. In 2.5, “The detailed Fermi levels (E_f), conduction band minimum (CBM), and valence band maximum (VBM) are shown in Figure 5c.” where the Figure 5c should be Figure 5d.

12. In 2.3, “A 0.1 wt% Pt cocatalyst was loaded on the photocatalyst via photodeposition.” The preparation process of Pt cocatalyst in the Supporting Texts should be provided.

Point-by-point Response to the Reviewers' Comments

Reviewer #1 (Remarks to the Author):

This work uses the hydrovoltaic effect to enhance the performance of photocatalytic water splitting. The idea is novel. Also, the system has good performance with H₂ and H₂O₂ production rates of 48.4 and 20.4 mmol g⁻¹ h⁻¹, respectively, and a high AQY of 56.2% at 400 nm, making it among the most efficient photocatalysts. Thus, I recommend the acceptance of this manuscript. However, the experiments and results are not clearly presented, and mechanism is not well explained, major revision is needed before publication, as noted below.

Response: We thank the referee for appreciating the impact and value of our study. We also appreciate the referee's constructive comments, which have helped us improve the quality of our manuscript. We have revised the manuscript accordingly.

1. It is not clear how the generated hydrovoltage can enhance the photocatalytic reaction. The hydrovoltage is established between two electrodes along the film, but in Fig. 5, this voltage seems to be applied to CoO/NC interface, there is no direct relation between them.

2. What is the direction of steam flow? Based on Fig. 5, the flow is along the width of the film. If this is the case, how hydrovoltage is created because two electrodes are symmetric and there should be no voltage difference.

Response: The questions 1 and 2 can be answered together. We have defined the direction of the water steam transport vertically through the PAA/CoO-NC film. The water steam is exposed on the surface of PAA/CoO-NC and then diffused along the nanochannel to the bottom. **Figure R1a** shows the photo of PAA/CoO-NC hydroelectric generator. The PAA/CoO-NC is placed in a closed reactor, in which the wetted Ar gas (Ar/H₂O) is brought, and a certain ambient humidity is formed in the reactor.

The direction of steam flow and produced hydrovoltage effect: **Figure R1b** shows the schematic of water steam diffusion on the PAA/CoO-NC film. When PAA/CoO-NC film exposed on water steam environment, the water steam enables H⁺ ions to diffuse from the surface to the bottom of PAA/CoO-NC due to its porous structure, which is similar to previous literature (Adv. Mater. 2018, 30, 1705925). Thus, the water steam flow diffused along the nanochannel to the bottom. The hydrovoltaic voltage is generated because of a gradient in the concentration of water molecules along the nanochannels from the top surface to the bottom of the film. The water steam diffusion on the PAA/CoO-NC film of a nanochannel based on the electrokinetic effect is illustrated in **Figure R1c, d**.

The PAA/CoO-NC generates a pressure-driven flow carriers counter-ions of H^+ to form an electric current in the flow, eventually reaching an equilibrium of H^+ ions diffusion, resulting in a constant voltage output.

How hydrovoltaic enhanced photocatalytic reaction: The photocatalytic reaction has been enhanced by generated hydrovoltaic effect based on two sides: (1) The hydrovoltaic effect induces an electric field in photocatalyst of CoO-NC and generates a strong interaction between the H^+ carriers and reaction centers of the nanostructure, thereby further improving the kinetics of water splitting (**Figure R1d**). (2) The height of the Schottky barrier between CoO and the NC interface decreases due to the hydrovoltaic effect, which significantly promotes carrier separation and transfer on the photocatalyst (**Figure 5f**). These features collectively enhance the photocatalytic performance of the proposed system. The hydrovoltaic-enhanced photocatalytic mechanism has already been discussed in the page 11 of main text.

Figure R1. (a) Photo of the PAA/CoO-NC hydroelectric generator for electricity generation under light illumination (AM 1.5G, 100 mW cm^{-2}) with Ar/H₂O injection; (b) The schematic of Ar/H₂O steam diffusion from the PAA/CoO-NC surface to the bottom of the film; (c) Schematic illustration of classical electrokinetic effect in the nanochannel of PAA/CoO-NC with Ar/H₂O steam flow and the ions diffusion along the negatively charged walls of PAA/CoO-NC nanochannels; (d) Equilibrium of Ar/H₂O steam and ions diffusion in a nanochannel of PAA/CoO-NC.

Device schematic and configuration used for electricity measurements: The schematic of device electrode has been redrawn as shown in **Figure R2**. The PAA/CoO-NC film is sandwiched with copper sheet as bottom electrode and copper mesh as top electrode, ensuring water steam permeated and electrons transferred in the vertical direction along with water steam diffusion.

Figure R2. Device schematic and configuration used for electricity measurements.

In the revised version, **Figure R1** is added as new **Figure S13** in the supporting information. **Figure R2** is added as new **Figure S1** in the supporting information. The corresponding photo and schematic of PAA/CoO-NC have been updated in **Figure 1b, c**. The relative discussion has been added in the page 7 of main text, page 18 of supporting information, and page 6 of supporting information and copied below:

*“The PAA/CoO-NC hydroelectric generator has been successfully constructed based on a moisture electrokinetic effect in a nanochannel of PAA/CoO-NC. ^[15, 30] The water steam is exposed on the surface of PAA/CoO-NC and then diffused along the nanochannel to the bottom due to its porous structure (the detail mechanism and water steam diffusion path seen in **Figure S13**). The PAA/CoO-NC generates a pressure-driven flow carriers counter-ions of H^+ to form an electric current in the flow, eventually reaching an equilibrium of H^+ ions diffusion, resulting in a constant voltage output. ^[15, 30]” (Page 7 of main text)*

*“**Figure S13a** shows the photo of PAA/CoO-NC hydroelectric generator. **Figure S13b** shows the schematic of water steam diffusion on PAA/CoO-NC film. When water steam is exposed on the surface of PAA/CoO-NC film, the water molecules enable mobility of H^+ ions diffused from surface to the bottom of PAA/CoO-NC along with its porous structure. The diffusion direction of water steam is perpendicular to the membrane downward, which is similar to previous literature ^[25]. The hydrovoltaic voltage is generated because of a gradient in the concentration of water molecules along the nanochannels from the top surface to the bottom of film as illustrated in **Figure S13c, d**. The PAA/CoO-NC generates the double layer overlaps and a pressure-driven flow carriers counter-ions of H^+ to form an electric current in the flow, eventually reaching an equilibrium of H^+ ions diffusion, resulting in a constant electrokinetic voltage” (Page 18 of supporting information)*

*“As shown in **Figure S1**...then pasted the copper sheet as the bottom electrode...Then the copper mesh electrode was pasted by epoxy slurry on the PAA/CoO-NC surface as the electrode, allowing the water steam to pass through.” (Page 6 of supporting information)*

3. Why H_2O_2 is generated instead of O_2 ?

Response: The photocatalytic product in our system is H_2O_2 rather than O_2 , which is proved through

the DFT-based calculations. As shown in **Figure R3a**, the photocatalyst CoO-NC shows a much lower free energy barrier of 0.37 eV for the H₂O₂ production process in the hydrovoltaic electric field (U=0.4 V) than the intrinsic barrier of 2.05 eV, indicating the hydrovoltaic effect is beneficial for the H₂O₂ reaction in PAA/CoO-NC system. For O₂ production path, the photocatalyst CoO-NC shows free energy barriers of 2.07 and 2.64 eV with or without hydrovoltaic effect, as shown in **Figure R3b**, much higher than the limiting step barrier for H₂O₂ generation. As a consequence, the DFT-based calculations indicate that the CoO-NC photocatalyst prefers to generate H₂O₂ rather than O₂ evolution.

Figure R3. (a) DFT calculation of free energy diagram for the three-step H₂O₂ production process on CoO-NC with (U=0.4 V) or without (U=0 V) hydrovoltaic electric field and the corresponding adsorption geometries structures on CoO-NC; (b) DFT calculation of free energy diagram for O₂ production process on CoO-NC with (U=0.4 V) or without (U=0 V) hydrovoltaic electric field and the corresponding adsorption geometries structures on CoO-NC.

In the revised version, **Figure R3a and 3b** are added as new **Figure S27** in the supporting information, and the relative discussion has been added in the page 11 of main text, and page 38 of supporting information and copied below:

“Density functional theory (DFT) based calculations further indicate that the CoO-NC photocatalyst prefers to generate H₂O₂ rather than O₂ evolution in the system (Figure S27).” (Page 11 of main text, detail see DFT calculations of supporting information)

4. The schematic in Fig. 5 is based on water, but most experiments were performed in steam, will the mechanism stay the same?

Response: I am sorry for the mistake. The word “water” in schematic of **Figure 5h** should be “steam”. we have checked and corrected the relevant annotations of water to Ar/H₂O steam in **Figure 5h** of page 20 in main text.

5. What is the condition of steam? Temperature? Is there any water droplets? According to the phase diagram (*J. Phase Equilib.*, 24 [3] 212-227 (2003)), if the temperature is high, CoO will convert to Co₃O₄. Thus temperature is important.

Response: Thanks very much for the valuable comments. The condition of steam is Ar/H₂O, the temperature of Ar/H₂O steam in the hydrovoltaic system is about 40 °C, and there are no water droplets (**Figure R4**), which is added as **Figure S10a** in the revised version. The CoO will not convert to Co₃O₄ because the temperature in hydrovoltaic system is lower than the CoO phase transformation temperature (2103 K from Ref. *J. Phase Equilib.*, 24 [3] 212-227 (2003)).

Figure R4. The detected temperature on PAA/CoO-NC surface over time with light illumination (light intensity: AM 1.5G, 100 mW cm⁻²) and at Ar/H₂O injection rate of 1100 ml h⁻¹.

6. What determines the flow velocity of steam or water? If it is determined by external pressure, then it should not depend on the illumination as in section 2.2, the authors sated "light illumination can increase the moving velocity".

Response: Thanks very much for the valuable comments. The moisture of Ar/H₂O, acting as an external force, affects the diffusion of water, which in turn creates the relative motion of water molecules and nanochannel of PAA/CoO-NC. The light illumination induces a higher temperature on the surface of PAA/CoO-NC as shown in **Figure R4**, bringing an increased moving velocity of water steam in the nanochannel of PAA/CoO-NC.

The relation between elevated temperature induced by light illumination and water steam velocity can be mathematically described as the following equations of diffusion coefficient equations R1. (Small 2018, 14, 1704473; Angew. Chem. Int. Ed. 2016, 55, 8003):

$$D=D_0e^{E_a/RT} \quad (R1)$$

where D , D_0 , E_a , e , R and T represent the diffusion coefficient of protons, the maximum diffusion coefficient at infinite temperature, activated energy, unit electric charge, gas constant and temperature, respectively. As the temperature increases of the surface by light irradiation, the thermal motion of molecules and the collisions between molecules are intensified, and the energy E_a is greatly increased, thus increasing the diffusion coefficient D according to equation R1.

In the revised version, the equation of R1 is added as new equation 9 in **Note S2** on page 13 of supporting information; **Figure R4** is added as new **Figure S10a**. The relative discussion has been added in the page 6 of main text, and page 13 of supporting information and copied below:

“Then, we investigated the influence of light illumination on the system for moisture moving. The light illumination induces a higher temperature on the surface of PAA/CoO-NC, bringing an increased moving velocity of water steam in the nanochannel of PAA/CoO-NC” (Page 6 of main text)

*“**Note S2: Factors of voltage tuning for light illumination on PAA/CoO-NC....** As the temperature increases of the surface, the thermal motion of molecules and the collisions between molecules are intensified, and the activation energy E_a is greatly increased, so is the diffusion coefficient D , thus increasing the output voltage according to equation (9) and (10). As a result, the light illumination induces a higher temperature on the surface of PAA/CoO-NC, bringing an increased diffusion and moving velocity of water steam in the nanochannel of PAA/CoO-NC...” (Page 13 of supporting information)*

7. In the stability test, does PAA/CoO-NC stay the same after four cycles of 20 h reaction? What will happen if prolong the reaction time?

Response: We investigated photocatalytic performance for a longer reaction period of 80 h. The performance retained 92% of its initial activity for PAA/CoO-NC with a slight degradation after a longer reaction period of 80 h, as shown in **Figure R5**. In the revised version, **Figure R5** is added as new **Figure S19** in the page 28 of supporting information, and the relative discussion has been added in the page 8 of main text

Figure R5. Cycling of photocatalytic hydrogen evolution over PAA/CoO-NC hydrovoltaic generator for a reaction period of 80 h.

Reviewer #2 (Remarks to the Author):

*The present manuscript reports on Hydrovoltaic effect-enhanced photocatalysis by PAA/CoO–NC system for efficient photocatalytic water splitting. The materials are tested as photocatalysts for H₂ evolution. Due to its optical properties, PAA/CoO–NC system is an interesting material to explore for photocatalytic application. **The manuscript in my view presents a novelty to be published in Nature communication.** However, some critical issues need to be addressed and manuscript revised prior to acceptance. **For these reasons, I recommend the publication of the present manuscript in Nature communication.** Major critical points are outlined in the following:*

Response: We thank the referee for appreciating the impact and value of our study. We also appreciate the referee's constructive comments, which have helped us improve the quality of our manuscript. We have revised the manuscript accordingly.

1. (a) *The increased photocatalytic performance upon modification with CoO is certainly not a surprise as it serves as a co-catalyst or a trap center for the material prepared. (b) Moreover, the manuscript omits some relevant information and review on hydrovoltaic effect as it related to photocatalysis check this article and reference it. (Zhong, T., Li, H., Zhao, T., Guan, H., Xing, L., & Xue, X. (2020). Self-powered/self-cleaned atmosphere monitoring system from combining hydrovoltaic, gas sensing and photocatalytic effects of TiO₂ nanoparticles. Journal of Materials Science & Technology. doi:10.1016/j.jmst.2020.11.002)*

Response: Thanks very much for the valuable comments. The questions can be divided two parts. Regarding question (b), we have cited the recommended paper in reference [17] as a review on hydrovoltaic effect on photocatalysis in the page 13 of main text.

Regarding question (a), the novelty of the work is that we proposed a hydrovoltaic effect-enhanced photocatalytic system. Generally, the CoO nanomaterial serves as a co-catalyst or a photogenerated holes trap center for enhanced photocatalytic performance. However, CoO itself can also act as a main photocatalyst for photocatalytic water splitting as previous literatures reported (Nat. Nanotechnol. 2014, 9, 69; Nat. Commun. 2021, 12, 1343). In our constructed hydrovoltaic generator system of PAA/CoO-NC, the CoO-NC forms an intimate heterostructure, in which CoO was used as a main photocatalyst and NC as a cocatalyst based on the matching energy band structure, achieving an efficient hydrovoltaic effect enhanced photocatalytic system. Other photocatalyst, such as TiO₂ can also be used in the hydrovoltaic effect-enhanced photocatalytic system, demonstrating the universality of the system, which has been discussed in Question 3 of Reviewer 3.

2. Carbon is known to possess large surface area; however, the work is silent about the effect of CoO on the surface area of the system.

Response: Thanks very much for the valuable comments. From the N₂ adsorption-desorption isotherm plots, the specific surface area of NC (**Figure R6**) is 61 m² g⁻¹. After the CoO was introduced, the specific surface area of CoO-NC is 73 m² g⁻¹, showing a slight difference. And the pore size distribution of both samples features typical mesoporous characteristics. In the revised version, **Figure R6** is added as new **Figure S6** in the page 9 of supporting information.

Figure R6. (a) N₂ adsorption-desorption isotherm of NC, CoO-NC and PAA/CoO-NC nanomaterials; (b) The corresponding pore size distribution curve of NC, CoO-NC and PAA/CoO-NC nanomaterials.

3. The specific wavelength of the light source used in the photocatalytic generation of H₂ is not state, even though Figure S16 depicts several wavelengths was used to determine AQY.

Response: The H₂ generation tests were conducted in the PAA/CoO-NC generator using a 300 W Xenon lamp irradiation with $\lambda > 300$ nm (PLS-SXE300, Beijing Perfectlight Technology Co., Ltd, 300 mW cm⁻²). The dependence of different wavelengths for H₂ generation in PAA/CoO-NC system was also estimated by various band-pass filters with different wavelengths of 400, 420, 425, 450, 500, 550, and 650 nm as shown in **Table R1**.

Table R1. Dependence of different wavelengths for H₂ generation in PAA/CoO-NC system and AQY calculation

Wavelengths (λ , nm)	H ₂ evolved (mmol h ⁻¹)	Light power (mW)	AQY (%)
400	2.12	6.27	56.2
420	2.06	6.01	54.2
425	2.02	5.95	53.1
450	1.93	5.50	51.8
500	1.84	4.97	49.2
550	1.51	3.73	48.9
650	0.63	1.39	46.3

AQY calculation: Take AQY@400 nm of PAA/CoO-NC as an example:

$$N = \frac{S \times P \times t \times \lambda}{h \times c} = \frac{6.27 \times 10^{-3} \times 400 \times 10^{-9} \times 3600}{6.626 \times 10^{-34} \times 3 \times 10^8} = 0.454 \times 10^{20}$$
$$AQY = \frac{2 \times n_{H_2} \times N_A}{N} \times 100\%$$
$$= \frac{6.02 \times 10^{23} \times 2.12 \times 10^{-3} \times 2}{0.454 \times 10^{20}} \times 100\% = 56.2\%$$

In the revised version, **Table R1** is added as new **Table S4**, and the relative discussion has been added in the page 26 of supporting information of **Note S6** and copied below:

*“Note S6. Calculation of the apparent quantum efficiency (AQY). The H₂ generation tests were conducted in the PAA/CoO-NC generator using a 300 W Xenon lamp irradiation with $\lambda > 300$ nm (PLS-SXE300, Beijing Perfectlight Technology Co., Ltd, 300 mW cm⁻²). The dependence of different wavelengths for H₂ generation and AQY calculations by various band-pass filters in PAA/CoO-NC system were shown in **Table S4**.” (Page 26 of supporting information)*

4. I am surprised that the effect of calcination at 600 °C on the surface of the CoO-NC was not reported, particularly if one considers that CoO-NC was annealed at 600 °C for 1 h, which might cause sintering and a consequent reduction of the surface area. In other words, one expects the bare CoO-NC to feature a higher specific surface area.

Response: Thanks very much for the valuable comments. The purpose of successive two-step annealing at 600 °C and 200 °C is to ensure the CoO-NC structure successfully synthesized although sintering-induced reduction of the surface area is inevitable. Annealing at 600 °C in the first step ensures a through decomposition of ZIF-67 and good crystallinity of Co nanoparticles according to a previous literature (J. Am. Chem. Soc., 139, 2017, 14143). Then, the obtained annealed sample was treated in a muffle furnace at 200 °C for 30 min to obtain CoO-NC as previous reports (Nano Energy, 61, 2019, 245).

In the revised version, the relative discussion has been added in the page 2 of supporting information and copied below:

“The purpose of successive two-step annealing at 600 °C and 200 °C is to ensure the CoO-NC structure successfully synthesized although sintering-induced reduction of the surface area is inevitable. Annealing at 600 °C in the first step ensures a through decomposition of ZIF-67 and good crystallinity of Co nanoparticles; and then treated in a muffle furnace at 200 °C to obtain CoO-NC [2, 3].” (Page 2 of supporting information)

5. If the DRS UV/VIS of both PAA/CoO and CoO-NC shows similar optical properties as presented in Fig S14. What is responsible for the difference in rate of H₂ evolution in both systems Because the difference is bandgap is not obvious.

Response: The photocatalytic H₂ evolution activity for PAA/CoO-NC and CoO-NC showed a big difference due to the different hydrovoltaic electric field induced in the systems. The generated hydrovoltaic electric field in PAA/CoO-NC is much stronger than that in the CoO-NC, resulting in a promoted carrier separation and transfer on the photocatalyst and higher kinetics of photocatalytic H₂ evolution as proved in the main text of **Figure 2**.

6. The mechanism presented in section 2.5 did not explain the pathway to the formation of H₂O₂ because in water splitting H₂ and O₂ are normally the by-product (Theoretically).

Response: Thanks very much for the valuable comments. We added the pathway to the formation of H₂O₂ and stated the reason why the by-product is H₂ and H₂O₂, not H₂ and O₂, by DFT calculation.

Figure R7. (a) DFT calculation of free energy diagram for the three-step H₂O₂ production process on CoO-NC with (U=0.4 V) or without (U=0 V) hydrovoltaic electric field and the corresponding adsorption geometries structures on CoO-NC; (b) DFT calculation of free energy diagram for O₂ production process on CoO-NC with (U=0.4 V) or without (U=0 V) hydrovoltaic electric field and the corresponding adsorption geometries structures on CoO-NC.

As shown in **Figure R7a**, the photocatalyst CoO-NC shows a much lower free energy barrier of 0.37 eV for the H₂O₂ production process in the hydrovoltaic electric field (U=0.4 V) than the intrinsic barrier of 2.05 eV, indicating the hydrovoltaic effect is beneficial for the H₂O₂ reaction in PAA/CoO-NC system. The photocatalyst CoO-NC shows free energy barriers of 2.07 and 2.64 eV for O₂

production with or without hydrovoltaic effect as shown in **Figure R7b**, much higher than the limiting step barrier for H₂O₂ generation. As a consequence, the photocatalyst of CoO-NC prefers to generate H₂O₂ rather than O₂ evolution.

In the revised version, **Figure R7a and 7b** are added as new **Figure S27** in the supporting information, and the relative discussion has been added in the page 11 of main text and page 38 of supporting information and copied below:

“Density functional theory (DFT) based calculations further indicate that the CoO-NC photocatalyst prefers to generate H₂O₂ rather than O₂ evolution in the system. (Figure S27).” (Page 11 of main text, detail see DFT calculations in supporting information)

7. The XPS results in Figure 1j and S3 did not explain if the CoO is decorated on the surface of the materials or found within the matrix, also the actual amount of the CoO as determined by the XPS is not reported.

Response: Thanks very much for the valuable comments. SEM and TEM images in **Figure 1** show that the CoO-NC nanoparticles are wrapped by the cross-linked PAA chains and distributed in the entire matrix. The XPS fits of Co 2p, N 1s, and O 1s in CoO-NC and PAA/CoO-NC are listed in **Table R2, R3** and **R4**, respectively. The fitted CoO content percentage in CoO-NC and PAA/CoO-NC is about 41% and 23%, respectively. In the revised version, **Table R2, R3** and **R4** are added as new **Table S1, S2** and **S3** in the supporting information, respectively, and the fitted results have been added in the revised version.

Table R2. Co 2p XPS fitting data of CoO-NC and PAA/CoO-NC

Catalysts	Co 2p _{3/2}			Co 2p _{1/2}		
	Peak (eV)	FWHM	Area	Peak (eV)	FWHM	Area
CoO-NC	778.6	1.01	23098	794.1	1.08	10765
PAA/CoO-NC	777.6	1.06	19852	793.7	1.02	7894

Table R3. N 1s XPS fitting data of CoO-NC and PAA/CoO-NC

Catalysts	Pyridinic N			Co-N			Pyrrolic N		
	Peak	FWHM	Area	Peak	FWHM	Area	Peak	FWHM	Area
CoO-NC	398.7	1.03	35687	399.4	1.02	34672	401.2	1.09	12456

PAA/CoO-NC	398.6	1.07	37789	399.4	1.01	18558	401.1	1.03	11098
------------	-------	------	-------	-------	------	-------	-------	------	-------

Table R4. O 1s XPS fitting data of CoO-NC and PAA/CoO-NC

Catalysts	Lattice oxygen			Hydroxyl		
	Peak (eV)	FWHM	Area	Peak (eV)	FWHM	Area
CoO-NC	530.3	1.03	22048	532.1	1.05	6765
PAA/CoO-NC	530.3	1.02	19842	531.9	1.02	16894

8. *The conclusion stated that the hydrovoltaic effect improves the hydrogen binding energy in the formation of active hydrogen during photocatalytic water splitting, resulting in excellent charge separation and transfer and accelerated reaction kinetics. However, the result discussion does not support this claim.*

Response: Thanks very much for the valuable comments. We have re-strengthened the elaboration of the conclusion in the revised text.

“..., improving the kinetics of water splitting and electron transport. ...”. (Page 11 of main text)

Reviewer #3 (Remarks to the Author):

*In this manuscript, Xu et al. proposed a hydrovoltaic effect-enhanced photocatalytic system in which an enhanced hydrovoltaic effect was achieved using polyacrylic acid (PAA) and cobaltous oxide (CoO)–nitrogen doped carbon (NC) and CoO–NC acted as a photocatalyst to generate H₂ and H₂O₂ products simultaneously. The hydrovoltaic effect was shown to decrease the Schottky barrier height between CoO and the NC interface and accelerate the kinetics of water splitting, thereby resulting in photocatalytic performance with high H₂ and H₂O₂ production rates. **This work realized a synergetic combination of the conventional photocatalysis and the newly emerged hydrovoltaic effect and may open a new sub-direction for sustainable development of clean energy.** However, before I can recommend its publication, the authors have to fully address following issues:*

Response: We thank the referee for appreciating the impact and value of our study. We also appreciate the referee's constructive comments, which have helped us improve the quality of our manuscript. We have revised the manuscript accordingly.

1. *In the introduction, “Hydrovoltaic technology is a renewable energy harvesting method that uses moisture interactions with nanostructured carbon materials”. The definition is not accurate. Hydrovoltaic technology can not only harvest energy from ambient moisture, but also from evaporation, raindrops, waves and so on. It is characterized by energy conversation based on the interaction between water and functional materials. The authors should find in which paper this terminology was first proposed.*

Response: Thanks very much for the valuable comments. In the revised version, the relevant literatures have been supplemented as Refs. [18-22] (Annu. Rev. Fluid Mech. 2004, 36, 381; Nat. Commun. 2014, 5, 3582; Nat. Nanotechnol. 2014, 9, 378; Phys. Rev. Lett. 2001, 86, 131; Nat. Nanotechnol., 12, 2017, 317); and the definition has been revised and copied below:

“Hydrovoltaic technology is a renewable energy harvesting method that can directly generate electricity by nanostructured materials interacting with water” (Page 2 of main text)

2. *In section 2.2, the influence of light illumination on the moisture moving velocity was investigated. An increased velocity results in an enhanced voltage generation. The author should clarify whether the enhancement is contributed solely by the increased moving velocity, any additional contribution from other factors, such as the photogenerated carriers or inhomogeneous distribution of the heat induced by light?*

Response: We agreed with the reviewer about the comments. In addition to an increased velocity of water steam induced by the light illumination, the inhomogeneous distribution of the heat (i.e. thermo-

electric effect) and photogenerated carriers (i.e. photoelectric effect) induced by light illumination have effects on an enhanced voltage generation.

Figure R8. Measured I-t characteristics of PAA/CoO-NC film for different testing conditions.

As shown in **Figure R8**, the photocurrent measurement of PAA/CoO-NC film was taken to detect the photogenerated current by using Keithly 2400 source meter under three testing conditions of the device. (i) The PAA/CoO-NC was tested in water to avoid hydrovoltaic effect and thermo-electric effect. The obtained transient current is supposed to be the photoelectric effect induced photocurrent. As soon as the light is turned on, the transient current induced by the photogenerated increases instantaneously to about 8 μA , indicating the photogenerated carriers' motion does exist in the system, which affects on the voltage output; (ii) The PAA/CoO-NC was tested in the ambient environment without water steam introduction to retaining its thermo-electric effect induced by the inhomogeneous distribution of the heat, and photoelectric effect induced by the photogenerated carriers. The transient current increases to about 25 μA with light illumination, indicating the thermo-electric effect exists in the system under light illumination; (iii) The PAA/CoO-NC was tested in the water steam environment with light illumination in case of the presence of photoelectric, thermo-electric, and hydrovoltaic effects. The transient current showed a maximum of about 150 μA , indicating an enhanced effect induced by hydrovoltaic effect. To sum up, the light illumination on the surface of PAA/CoO-NC induced an elevated temperature on the surface and resulted in the moving velocity of water steam (hydrovoltaic effect), photogenerated carriers' motion, and thermo-electric effect, which collectively contributed to a higher output voltage.

In the revised version, **Figure R8** has been added as new **Figure S10b** and the relative discussion has been added in the page 6 of main text, and page 13 of supporting information of **Note S2** and copied below:

“In addition to an increased velocity of water steam induced by the light illumination, the inhomogeneous distribution of the heat (i.e. thermo-electric effect) and photogenerated carriers (i.e. photoelectric effect) induced by light illumination have effects on an enhanced voltage generation (detailed relation seen in Supporting Information of Note S2, Figure S10). (Page 6 of main text)

“In addition to an increased velocity of water steam induced by the light illumination contributes to an increased voltage output. The thermo-electric effect and photoelectric effect induced by light illumination have effects on enhanced voltage generation. The photocurrent measurement of PAA/CoO-NC film was taken to detect the photogenerated current by using Keithly 2400 source meter under three testing conditions of the device. The three conditions are tested below; (i) The PAA/CoO-NC was tested in water to avoid hydrovoltaic effect and thermo-electric effect. The obtained transient current is supposed to be the photoelectric effect induced photocurrent. As soon as the light is turned on, the transient current induced by photogenerated increases instantaneously to about 8 μA , indicating the photogenerated carriers' motion does exist in the system, which has an effect on the voltage output. (ii) The PAA/CoO-NC was tested in the ambient environment without water steam introduction to retaining its thermo-electric effect. The transient current increases to about 25 μA with light illumination, indicating the thermo-electric effect exists in the system under light illumination; (iii) The PAA/CoO-NC was tested in the water steam environment with light illumination in case of the presence of photoelectric, thermo-electric, and hydrovoltaic effects. The transient current showed a maximum of about 150 μA , indicating an enhanced effect induced by hydrovoltaic effect.”

(Page 13 of supporting information of Note S2)

3. In the system, CoO was used as the active centers for water splitting and was shown to have a desirable band alignment with NC groups. I'm wondering if such a strategy of materials design can be generalized to a range of metal oxides, notably TiO_2 that is mostly used for photocatalysis. The authors are suggested to perform additional experiments to demonstrate the generality.

Response: Thanks very much for the valuable comments. We introduced TiO_2 into our system to demonstrate the generality. We prepared PAA/ TiO_2 -NC hybrid (**Figure R9a-c**). The energy band structure of PAA/ TiO_2 -NC based on UV-vis DRS and UPS analysis is obtained (**Figure R9d-f**). We investigated the electricity generation of PAA/ TiO_2 -NC generator and its enhanced photocatalytic performance. As shown in **Figure R9g**, PAA/ TiO_2 -NC reaches the highest voltage (~ 345 mV) at a 1100 ml h^{-1} Ar/ H_2O steam injection rate with light illumination, confirming the generation of hydrovoltaic effect in PAA/ TiO_2 -NC generator. The H_2 evolution amount of PAA/ TiO_2 -NC nanogenerator is 1.2 mmol at the Ar/ H_2O steam injection rate of 1100 ml h^{-1} , and O_2 gas is detected to

be 0.5 μmol in the first cycle as shown in **Figure R9h**, higher than the performance of PAA/TiO₂-NC in bulk water (**Figure R9i**), suggesting enhanced photocatalytic water splitting performance in the system of PAA/TiO₂-NC induced by the hydrovoltaic effect.

Figure R9. (a) TEM image of PAA/TiO₂-NC; (b) HRTEM image of PAA/TiO₂-NC; (c) XRD pattern of PAA/TiO₂-NC; (d) UV-vis DRS spectrum and Kubelka-Munk function vs. the energy of incident light plot for TiO₂; (e) UPS spectrum of the valence band region of TiO₂ and UPS spectrum of the cutoff region for TiO₂; (f) The corresponding energy band structures for the materials; (g) Measured output voltage for PAA/TiO₂-NC over time with light illumination (light intensity: AM 1.5G; 100 mW cm⁻²); (h) Time-dependent photocatalytic H₂ and O₂ production of PAA/TiO₂-NC at different Ar/H₂O rate at 1100 ml h⁻¹, and submerged in water; Pt cocatalyst is loaded using a photodeposition method; The light source is a solar simulator at AM 1.5G illumination (100 mW cm⁻²); (i) The corresponding time-dependent photocatalytic H₂ and O₂ production submerged in water.

In the revised version, **Figure R9** is added as new **Figure S21** in the supporting information, and the relative discussion has been added in the page 8 of the main text and in page 30 of the supporting information and copied below:

*“The PAA/TiO₂-NC photocatalyst has also prepared to further demonstrate the generality of hydrovoltaic effect enhanced photocatalysis as shown in **Figure S21**.” (Page 8 of main text)*

“...The PAA/TiO₂-NC reaches a voltage (~345 mV) at a 1100 ml h⁻¹ Ar/H₂O steam injection rate with light

illumination, confirming the generation of hydrovoltaic effect in PAA/TiO₂-NC generator (**Figure S21g**). The H₂ evolution amount of PAA/TiO₂-NC nanogenerator is 1.2 mmol at the Ar/H₂O steam injection rate of 1100 ml h⁻¹, and O₂ gas is detected to be 0.5 mmol in the first cycle as shown in **Figure S21h**, higher than the performance of PAA/TiO₂-NC in bulk water (**Figure S21i**), suggesting the generality of hydrovoltaic effect enhanced photocatalysis” (Page 30 of supporting information)

4. The generated hydrovoltaic electricity is by pumping moisture steam through the materials, which costs extra energy and may result in an overall extremely low efficiency. Actually, hydrovoltaic technology enables electricity harvesting from natural evaporation which does not any extra energy input. In this regard, can the authors employ the evaporation-induced electricity to enhance the photocatalysis?

Response: Thanks very much for the valuable comments. The experimental for natural water-evaporation-induced hydrovoltaic effect in a PAA/CoO-NC sample has been conducted as illustrated in **Figure R10**, and it indeed has effect on photocatalysis.

Figure R10. (a) Photo of the PAA/CoO-NC hydroelectric generator for measuring water evaporation-induced voltage under light illumination (AM 1.5G, 100 mW cm⁻²); (b) Measured output voltage for PAA/CoO-NC over time with or without light illumination (light intensity: AM 1.5G; 100 mW cm⁻²); (c) Time-dependent photocatalytic H₂ and H₂O₂ production of PAA/CoO-NC; Pt cocatalyst is loaded using a photodeposition method; The light source is a solar simulator at AM 1.5G illumination (100 mW cm⁻²).

To simulate the natural evaporation, we designed a PAA/CoO-NC generator by inserting it into a 100 ml beaker with deionized water covering the bottom electrode under light illumination as shown in **Figure R10a**. An open-circuit voltage between the two electrodes is generated and gradually rises to 267 mV when the capillary water reaches its maximum height along the PAA/CoO-NC sheet in about 30 mins. The voltage output increases to 318 mV with light illumination at AM 1.5G illumination (100 mW cm⁻²) as shown in **Figure R10b**. We conducted H₂ production experiments to investigate water-evaporation-enhanced photocatalytic water splitting reactions. As shown in **Figure R10c**, the

H₂ evolution amount of PAA/CoO-NC nanogenerator is 8.3 mmol, and the oxidation product of H₂O₂ is measured to be 3.7 mmol, higher than the generalized powder photocatalyst system as shown in **Figure 3**, suggesting enhanced photocatalytic performance by the water-evaporation-induced hydrovoltaic effect.

In the revised version, **Figure R10** is added as new **Figure S20** in the supporting information, and the relative discussion has been added in the page 8 of main text and in page 29 of supporting information and copied below:

*“In addition, the hydrovoltaic effect generation and its enhanced photocatalysis are also demonstrated in a natural water-evaporation-induced hydrovoltaic system (**Figure S20**).” (Page 8 of main text)*

*“**Figure S20a** shows the photo of PAA/CoO-NC film for water-evaporation hydrovoltaic generation. As shown in **Figure S20b**, an open-circuit voltage between the two electrodes is generated and gradually rises to 267 mV. The voltage output increases to 318 mV with light illumination at AM 1.5G illumination (100 mW cm⁻²). As shown in **Figure S20c**, the H₂ evolution amount of PAA/CoO-NC nanogenerator is 8.3 mmol, and the oxidation product of H₂O₂ is measured to be 3.7 mmol, suggesting an enhanced photocatalytic performance by the water-evaporation-induced hydrovoltaic effect.” (Page 29 of supporting information)*

5. To further verify the photocatalytic effect of PAA/CoO-NC, the authors are suggested to carry out control experiments of photocatalytic H₂/H₂O₂ production and provide in situ Raman spectra of pure PAA membranes loaded with Pt cocatalyst at 1100 ml h⁻¹ Ar/H₂O steam injection.

Response: Thanks very much for the valuable comments. We conducted the in situ Raman spectra and photocatalytic performance on PAA/Pt at the 1100 ml h⁻¹ Ar/H₂O steam injection under light illumination to eliminate the catalytic activity of the substrate. As shown in **Figure R11a**, a peak centered at 2100 cm⁻¹ assigned to Pt–H vibration, which gradually increases in intensity over time. The redshift of the Pt–H peak is from 2097 to 2104 cm⁻¹, indicating a weak hydrovoltaic electric field. The O–H stretching observed in PAA/Pt evolves at approximately 3200 and 3400 cm⁻¹, which increases slowly in intensity over time, indicating a weak interaction between PAA/Pt and water molecules. The H₂ evolution amount of PAA/Pt membrane is 7.8 μmol, and a trace amount of H₂O₂ of 0.3 μmol is detected due to the introduced weak hydrovoltaic effect of PAA and intrinsic hydrogen production characteristic of Pt, indicating a slight influence of substrate for the system (**Figure R11b**).

Figure R11. (a) In situ Raman spectra of the PAA/Pt surface over time with hydrovoltaic effect under light irradiation; (b) Time-dependent photocatalytic H₂ and H₂O₂ production of PAA/Pt; Pt cocatalyst is loaded using a photodeposition method; The light source is a solar simulator at AM 1.5G illumination (100 mW cm⁻²).

In the revised version, **Figure R11** is added as **Figure S23** in the supporting information, and the relative discussion has been added in the page 10 of the main text and page 33 of the supporting information and copied below:

*“The in situ Raman spectra of PAA/Pt was also investigated in **Figure S23** to exclude the catalytic activity of the substrate, indicating a great enhancement on photocatalyst induced by hydrovoltaic effect.”* (Page 10 of main text)

*“We conducted photocatalytic performance measurement and measured the in situ Raman spectra on PAA/Pt at the 1100 ml h⁻¹ Ar/H₂O steam injection under light illumination. As shown in **Figure S23a**, a peak centered at 2100 cm⁻¹ assigned to Pt–H vibration is observed, which gradually increases in intensity over time. The slight redshift of the Pt–H peak from 2097 to 2104 cm⁻¹ and the slow increase in intensity O–H stretching at approximately 3200 and 3400 cm⁻¹ over time, indicate a weak hydrovoltaic electric field in PAA/Pt. The H₂ evolution amount of PAA/Pt membrane is 7.8 μmol, and trace amount of H₂O₂ of 0.3 μmol is detected due to the introduced hydrovoltaic effect of PAA and intrinsic hydrogen production characteristic of Pt (**Figure S23b**).”* (Page 33 of supporting information)

6. There are some misquotes in the introduction, such as “For example, a hydroelectric generator comprising the ionic polymer Nafion and a poly(*N*-isopropylacrylamide) hydrogel was developed to generate electricity [17, 18]”. Please check the literature comprehensively.

Response: Thanks very much for the valuable comments. We have checked the literature comprehensively, and the cited reference [17, 18] has been corrected to reference [33] Energy Environ. Sci. 15, 2489-2498 (2022).

7. In Figure S1, the description on the device fabrication process in figure legend does not match its schematic illustration. For example “Commercial epoxy slurry is used to paint two “L-shaped” with predesigned dimensions on the substrate (Step 1)”; here, the electrodes should be “L-shaped”, not

epoxy.

Response: Thanks very much for the valuable comments. The schematic of device electrode has been revised and more clearly represented as shown in **Figure R12**. In the revised version, it is added as new **Figure S1** in page 6 of Supporting Information.

Figure R12. Device schematic and configuration used for electricity measurements.

8. In Note S1, “relative humidity by pumping a mixture of dry and wet nitrogen”, while in the paper, the humidity is controlled by “Ar/H₂O steam injection”.

Response: Thanks very much for the valuable comments. We have revised the annotation to “argon gas” in **Note S1** in page 10 of supporting information.

9. In Figure S7 (c) and Figure S8 (b), “The increase in the current density may be attributed to the increased amount of water moving and diffusion as *W* increases”. While in general perception, the current increases with the width of the hydroelectric generator, and the current density is nearly constant. Please explain it.

Response: Thanks very much for the valuable comments. We re-tested the voltage and current density for PAA/CoO-NC and CoO-NC devices by taking the surface area and thickness as the major influencing factors. The device surface area is selected as 15 cm² and the thickness of film is selected to be 300 μm.

The PAA/CoO-NC and CoO-NC devices with sandwiched electrodes for electricity tests are shown in **Figure R13a**. The surface area and thickness are important factors for electricity generation. As the surface area of the device increases from 1 cm² to 18 cm², the voltage and current density for PAA/CoO-NC and CoO-NC all showed a slight decrease, which is due to the additional defects introduced into the devices as the surface area increases. The voltages are measured to be ≈280 mV and ≈100 mV for PAA/CoO-NC and CoO-NC devices, respectively, while the current density showed a basically stable value around ≈12 μA cm⁻² and ≈4 μA cm⁻² for PAA/CoO-NC and CoO-NC device, respectively (**Figure R13b, c**). These results indicate that the surface area of devices has little effect

on the electricity generation for the devices. Considering a larger surface area of the device is beneficial for photocatalysis due to complete exposure to light. We selected the device surface area of 15 cm² for the generation of electricity and its photocatalytic reactions.

For the effect of thickness exploration, we prepared different devices with thicknesses of 100, 300, 500 and 700 μm for electricity generation. The devices showed higher electricity generation at 300 μm for PAA/CoO-NC and CoO-NC (**Figure S8**).

Figure R13. (a) The diagram of devices (surface area and thickness as important factors); (b) Measurements of voltage and current density on PAA/CoO-NC with different surface areas. Δt = 500 s; (c) Measurements of voltage and current density on CoO-NC with different surface areas. Δt = 500 s; (d) The effect of different thicknesses on the voltage and current density measurements for PAA/CoO-NC in a closed reactor with 1100 ml h⁻¹ Ar/H₂O steam flow (area of 15 cm²); (e) The effect of different thicknesses on the voltage and current density measurements for PAA/CoO-NC in a closed reactor with 1100 ml h⁻¹ Ar/H₂O steam flow (area of 15 cm²).

In the revised version, **Figure R13a, b, and c** are added as new **Figure S7**, and the relative discussion has been added in the page 5 of main text and page 10 of supporting information and copied below:

*“First, we optimized the surface area, and thickness of the electricity generator, as illustrated in **Figures S7, 8** and **Note S1**. The optimal surface area, and thickness of the electricity generator are 15 cm^2 , and $300\text{ }\mu\text{m}$, respectively. (Page 5 of main text)*

*“The PAA/CoO-NC and CoO-NC devices with sandwiched electrodes for electricity tests are shown in **Figure S7a**. As the surface area of the device increased from 1 cm^2 to 18 cm^2 , the voltage and current density for PAA/CoO-NC and CoO-NC all showed a slight decrease, which is due to the additional defects introduced into the devices as the surface area increases. The voltages are measured to be ≈ 280 and $\approx 100\text{ mV}$ for PAA/CoO-NC and CoO-NC device, respectively, while the current density showed a basically stable value around ≈ 12 and $\approx 4\text{ }\mu\text{A cm}^{-2}$ for PAA/CoO-NC and CoO-NC device, respectively (**Figure S7b, c**). These results indicate that the surface area of devices has little effect on the electricity generation for the devices. Considering a larger surface area of the device is beneficial for photocatalysis due to complete exposure to light, we selected the device surface area of 15 cm^2 for the generation of electricity and its photocatalytic reactions.” (Page 10 of supporting information)*

10. In Figure 2a, the authors need to explain why the voltage at a 1300 ml h^{-1} Ar/H₂O steam injection rate is lower than that at 1100 ml h^{-1} . Does the liquid water cover the PAA/CoO-NC membrane?

Response: Thanks very much for the valuable comments. The water steam will completely saturate and cover the entire membrane at a higher humidity of Ar/H₂O steam 1300 ml h^{-1} , resulting in the disappearance of water steam gradient and hydrovoltaic effect (Adv. Sci. 9, 2022, 2201586). The explanation has been added in page 6 of the main text.

11. In 2.5, “The detailed Fermi levels (E_f), conduction band minimum (CBM), and valence band maximum (VBM) are shown in Figure 5c.” where the Figure 5c should be Figure 5d.

Response: Thanks very much for the valuable comments. We have corrected the mistake of **Figure 5d** in the page 10 of main text.

12. In 2.3, “A 0.1 wt% Pt cocatalyst was loaded on the photocatalyst via photodeposition.” The preparation process of Pt cocatalyst in the Supporting Texts should be provided.

Response: Thanks very much for the valuable comments. The Pt cocatalyst loading description of “In a typical procedure of photochemical loading Pt on PAA/CoO-NC photocatalyst, 50 mg PAA/CoO-NC nanomaterial and H₂PtCl₆ (4 mg mL⁻¹, 0.25 mL) were dispersed in an aqueous solution containing 20 mL H₂O. After the light treatment for 1 h, the sample was centrifuged and washed by deionized water twice and then free-dried. The Pt content relative to PAA/CoO-NC reached about 0.1 wt%” has been added in page 3 of supporting information

in the revised version.

REVIEWERS' COMMENTS

Reviewer #1 (Remarks to the Author):

The authors have addressed questions from all the reviewers very well, I recommend the acceptance of this manuscript.

Reviewer #2 (Remarks to the Author):

After a careful review of the revised manuscript and the rebuttal comment from the author, it was observed that the authors have addressed all the observation made in the original manuscript.